# EXPLOITING CLASS ACTIVATION VALUE FOR PARTIAL-LABEL LEARNING

**Fei Zhang**[1,2]   **Lei Feng**[3,†]   **Bo Han**[2,†]   **Tongliang Liu**[4]
**Gang Niu**[5]   **Tao Qin**[6]   **Masashi Sugiyama**[5,7]

[1]Shanghai Jiao Tong University   [2]Hong Kong Baptist University   [3]Chongqing University
[4]The University of Sydney   [5]RIKEN   [6]Microsoft Research Asia   [7]The University of Tokyo
ferenas@sjtu.edu.cn, lfeng@cqu.edu.cn, bhanml@comp.hkbu.edu.hk,
tongliang.liu@sydney.edu.au, gang.niu@riken.jp, taoqin@microsoft.com, sugi@k.u-tokyo.ac.jp

## ABSTRACT

*Partial-label learning* (PLL) solves the multi-class classification problem, where each training instance is assigned a set of candidate labels that include the true label. Recent advances showed that PLL can be compatible with deep neural networks, which achieved state-of-the-art performance. However, most of the existing deep PLL methods focus on designing proper training objectives under various assumptions on the collected data, which may limit their performance when the collected data cannot satisfy the adopted assumptions. In this paper, we propose to exploit the learned intrinsic representation of the model to identify the true label in the training process, which does not rely on any assumptions on the collected data. We make two key contributions. As the first contribution, we empirically show that the *class activation map* (CAM), a simple technique for discriminating the learning patterns of each class in images, could surprisingly be utilized to make accurate predictions on selecting the true label from candidate labels. Unfortunately, as CAM is confined to image inputs with convolutional neural networks, we are yet unable to directly leverage CAM to address the PLL problem with general inputs and models. Thus, as the second contribution, we propose the *class activation value* (CAV), which owns similar properties of CAM, while CAV is versatile in various types of inputs and models. Building upon CAV, we propose a novel method named CAV Learning (CAVL) that selects the true label by the class with the maximum CAV for model training. Extensive experiments on various datasets demonstrate that our proposed CAVL method achieves state-of-the-art performance.

## 1 INTRODUCTION

To liberate humans from exhaustive label annotation work, numerous researchers have dedicated themselves to investigating various *weakly supervised learning* (WSL) (Zhou, 2017) problems, including but not limited to noisy-label learning (Liu & Tao, 2015; Xia et al., 2020; Han et al., 2020), semi-supervised learning (Zhu & Goldberg, 2009; Miyato et al., 2018; Luo et al., 2018), and multiple-instance learning (Zhou et al., 2012). This paper focuses on another popular WSL problem called *partial-label learning* (PLL) (Jin & Ghahramani, 2002; Cour et al., 2011), which aims to learn a model from training examples equipped with a set of candidate labels that include the true label. Due to the cost and difficulty of annotating every example exactly with the true label in huge datasets, PLL has been widely applied to various tasks such as multimedia context analysis (Zeng et al., 2013) and web mining (Luo & Orabona, 2010).

To solve the PLL problem, there are two mainstream strategies to discriminate the unknown true label from candidate labels, including the *average-based strategy* (ABS) and the *identification-based strategy* (IBS). ABS always treats each candidate label equally and averages the model outputs of

---

†Correspondence to Lei Feng (lfeng@cqu.edu.cn) and Bo Han (bhanml@comp.hkbu.edu.hk).

all candidate labels for prediction (Hüllermeier & Beringer, 2006; Cour et al., 2011; Feng et al., 2020; Yao et al., 2020; Wen et al., 2021). IBS concentrates on iteratively selecting one label from the candidate label set as the true label to exclude the uncertainty, forming PLL into an ordinary classification problem (Jin & Ghahramani, 2002; Liu & Dietterich, 2014; Zhang & Yu, 2015; Zhang et al., 2016). Among these methods, only several of them (Yao et al., 2020; Wen et al., 2021) can be compatible with deep neural networks, which achieved state-of-the-art performance.

Most of the existing PLL methods aim at designing proper training objectives under various assumptions on the collected data. For example, the consistent methods proposed by Feng et al. (2020) are based on a specific data generation assumption. The derivation of the PRODEN method (Lv et al., 2020) stems from the assumption that the true label achieves the minimal loss among the candidate labels. Yan & Guo (2021) proposed the MGPLL method based on the assumption of non-random label noise. Although these methods have achieved generally great empirical performance, their performance could be degraded when the collected data cannot meet the adopted assumptions.

In this paper, we aim to investigate a novel PLL method by exploiting the learned intrinsic representation of the model to identify the ground truth in the training process without relying on any assumptions on the collected data. We focus on the *class activation map* (CAM) (Zhou et al., 2016), which is a prevailing tool for visualizing the representation information of *convolutional neural network* (CNN)-based models and could be easily obtained by the weighted linear sum of the feature maps. As CAM is able to discriminate the learning patterns of each class for the model, we conjecture that CAM could be used to guide the model to spot the true label. To verify this assumption, we conducted a pilot experiment on CIFAR-10 (Krizhevsky et al., 2009), where the candidate label sets were generated in two different ways. As shown in Figure 1 (please refer to Section 2.2), it is surprising to find that CAM can potentially help the model recognize the true labels, and such capacity of CAM synchronously changes with the classifier performance during the whole training phase. Based on the experimental results, it can be conjectured that such intrinsic representations can be useful to distinguish the true label from candidate labels for PLL.

Motivated by the above observations, we naturally consider leveraging CAM to solve the PLL problem. However, CAM was only proposed for addressing image datasets with using deep models built on CNN, revealing two limitations on its application. Firstly, CAM cannot be adopted to the classifier based on shallow models such as the linear model and *multilayer perceptron* (MLP). Secondly, CAM is unable to deal with any inputs other than the image. To overcome these shortcomings, we for the first time propose a simple but effective tool—the *class activation value* (CAV) to capture the learned representation information in a more general way, which is essentially the weighted output of the classifier. Experimental results in Figure 2 and Appendix B.7 also shows that our CAV works similarly as CAM during the training phase.

Building upon CAV, we propose a simple yet effective PLL method called CAV Learning (CAVL) that guides the model to differentiate the true label from the candidate set during the training process. Specifically, we first train the model by treating all candidate labels equally and obtain CAVs of given training examples, and then regard the class with the maximum CAV as the true label for each instance during the training process. In this way, CAVL transforms PLL into supervised learning, thereby making the model reliable to recognize the true label with learned representation information by CAV. Extensive experiments on benchmark-simulated and real-world datasets show that our proposed CAVL method achieves state-of-the-art performance.

## 2 DISCOVERING CLASS ACTIVATION MAP FOR PARTIAL-LABEL LEARNING

In this section, we provide a detailed discussion about our motivation and empirically show that the class activation map could be helpful for addressing PLL.

### 2.1 CLASS ACTIVATION MAP

The *class activation map* (CAM) (Zhou et al., 2016) is a popular and elementary mechanism in computer vision to represent the discriminative part of an input image captured by the classifier to identify each class. In other words, CAM manifests the learning patterns of the classifier, denoted as $f$, to the specific class in the image. Let us denote an input image as $x \in \mathbb{R}^{d=c \times h \times w}$ (with $c$ channels, height $h$ and width $w$) and CAM of $x \in \mathbb{R}^{d=c \times h \times w}$ as $m \in \mathbb{R}^{k \times h \times w}$ where $k$ is the

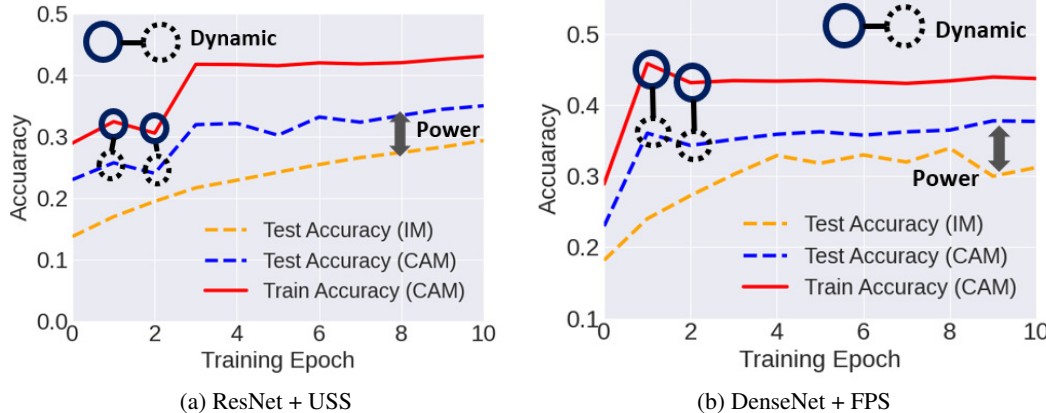

(a) ResNet + USS                    (b) DenseNet + FPS

Methods in (a) are implemented by ResNet with using USS to generate the candidate set. Methods in (b) are conducted by DenseNet with the candidate label sets produced by FPS. For three lines representing accuracy performance, the results drawn by the blue dashed line and red line ("CAM" line) are obtained by using the "CAM label" selection strategy, namely training the model with regarding the class with the most number of activated seeds in its CAM as the true label. Specifically, the red line depicts the training accuracy measured by selecting the CAM with the maximum activated seeds from the candidate set as the true label, and the blue line simply depicts the test accuracy of the classifier. The yellow dashed line ("IM" line) depicts the test accuracy of the classifier trained by IM. The paired circle indicates the similar accuracy fluctuation between the classifier and CAM, which shows the `dynamic` attribute. The double arrow marks the performance gap between "CAM" label selection strategy and IM, which represents the `power` attribute.

Figure 1: Comparison of accuracy performance of two different label selection strategies.

number of classes. We note that $m$ could be generated by training a specific classification model $f_c$, which normally comprises of an encoder function $e : \mathbb{X} \to \mathbb{R}^{c_f \times h \times w}$ for feature map extraction, a Global Average-Pooling layer $gap : \mathbb{R}^{c_f \times h \times w} \to \mathbb{R}^{c_f \times 1}$, and a linear layer with weight $\boldsymbol{\theta} \in \mathbb{R}^{k \times C_f}$, where $c_f$ is the number of channels in the feature maps. Therefore, CAM of $\boldsymbol{x}$ related to the $j$-th class could be obtained by

$$\boldsymbol{m}^j = \sum\nolimits_{i=1}^{C_f} (\boldsymbol{\theta}^j)_i e^i(\boldsymbol{x}), j \in \{1, \ldots, k\}, \tag{1}$$

where $\boldsymbol{\theta}^j \in \mathbb{R}^{1 \times C_f}$ is the corresponding linear weight related to the $j$-th object. CAM is a weighted linear sum of feature maps and the linear weights. The weakness of CAM is the limited applications, since not all classification network architectures follow the model $f_c$ mentioned above. To improve its generalization for any CNN-based model $f$, *Gradient-weighted CAM* (Grad-CAM) (Selvaraju et al., 2017) was proposed to implement such an internal representation by reasonably leveraging class-specific gradient information. Concretely, let us denote by $\boldsymbol{g}^j \in \mathbb{R}^{d=c \times h \times w}$ the Grad-CAM of $\boldsymbol{x}$ related to the $j$-th object, and $\boldsymbol{g}^j$ could be expressed as

$$\boldsymbol{g}^j = \frac{1}{h \times w} \sum\nolimits_{z=1}^{c_f} \sum\nolimits_{m=1}^{h} \sum\nolimits_{n=1}^{w} \frac{\partial f^j(\boldsymbol{x})}{\partial e^z(\boldsymbol{x})_{m,n}} e^z(\boldsymbol{x}). \tag{2}$$

Intuitively, the derivative of logit $f(\boldsymbol{x})$ with respect to feature map $e(\boldsymbol{x})$ is used as the weights for calculating $\boldsymbol{g}$. Note that both Grad-CAM and CAM are generated by weighted sums of the feature maps, and Grad-CAM is equivalent to CAM when $f$ follows the same architecture as $f_c$. Based on Grad-CAM, Grad-CAM++ (Chattopadhay et al., 2018) aims to provide better visual explanations of CNN and occurrences of multiple foreground objects. Thanks to such a simple technique, numerous problems in computer vision such as *interpretation* (Xu et al., 2019a) and *weakly supervised semantic segmentation* (Wei et al., 2016) have achieved marvelous progress. In this paper, we aim to extract and improve the useful knowledge from CAM to address the PLL problem, leading the classifier to identify the ground truth.

## 2.2 PILOT EXPERIMENT ON CAM

Generally, Grad-CAM or CAM is treated as the internal representation of $f$. We believe that such a mechanism could guide $f$ to differentiate the true label from the candidate set because it is constructed by taking advantage of internal elements in $f$. To validate our conjecture, we conducted a pilot experiment by adopting two different label selection methods. The first one is the *intuitive*

*method* (IM), which simply regards all the labels from the candidate sets as the true labels with using cross entropy loss. The second one is the "CAM label" selection strategy, which discriminates the true label from the candidate label set by using CAM. During each training epoch, we calculated the number of the foreground seeds of CAM from each candidate label set. Specifically, to calculate the number of foreground seeds of $m^j$, we counted the elements with positive values in $m^j$. Note that each $x$ could possess $k$ CAMs and the true label is always in the candidate set. Thus we selected the label from the candidate set, whose CAM owns most foreground seeds as the true label. The label candidate label sets were generated by two different approaches:*(I) Uniformly Sampling Strategy* (USS). Uniformly sampling the candidate label set for each training instance from all the possible candidate label sets (Feng et al., 2020). *(II) Flipping Probability Strategy* (FPS). By setting a flipping probability $q$ to any false label, the false label could be selected as a candidate label with a probability $q$ (Feng & An, 2019a; Yan & Guo, 2020; Lv et al., 2020; Wen et al., 2021). Here the detailed training settings of the experiments are similar to the experiment part (please refer to Section 4.1.1 for details). ResNet (He et al., 2016) and DenseNet (Huang et al., 2017) were chosen as the backbones to train a classifier on CIFAR-10 (Krizhevsky et al., 2009).

Figure 1 shows the average results of 5-time trials and we can find that CAM demonstrates two helpful features during the training phase. Firstly, it is clear that the performance of the classifier trained by "CAM label" selection strategy is better than IM (the gap between the blue dashed line and the yellow dashed line), which shows that CAM could potentially guide the classifier toward the true label from the candidate set. Here we name this attribute of CAM as `power`. Secondly, it is found that the accuracy of identifying true labels by CAM is synchronously changed with that by the classifier. Several fluctuations (marked by the paired circle) of the classifier performance (test accuracy) and CAM accuracy are similar to a large extent, resulting in continuous improvement to the model itself. Here we name this observed feature of CAM as `dynamic`. The `power` and `dynamic` attributes disclose the fact that *CAM may show more than we thought*, which is an inspiring finding to address PLL. The `power` attribute motivates us to consider that the classifier may learn more accurate true labels if it is forced to approximate the label distribution recognized by CAM. The `dynamic` attribute may guarantee that such guidance would be constantly effective during the whole training phase since CAM would synchronously be self-improved as the classifier becomes stronger. Hence it would be reasonable and meaningful to explore such internal sign as guidance for PLL.

## 3 PROPOSING CLASS ACTIVATION VALUE FOR PARTIAL-LABEL LEARNING

### 3.1 CLASS ACTIVATION VALUE—SIMPLE AND GENERAL REPRESENTATION

In this paper, we aim to propose a CAM-like tool as guidance for $f$ to identify the true label from candidate labels. CAM or Grad-CAM is specially designed for *convolutional neural network* (CNN)-based models, which mainly deals with image datasets with retaining spatial information. However, to the best of our knowledge, most PLL methods are implemented in datasets with various forms, including the image datasets such as MNIST (LeCun et al., 1998), and feature vectors such as Bird-Song (Briggs et al., 2012). The networks on these datasets could be directly adapted to elementary models such as the linear or *multilayer perceptron* (MLP) for $f$ to deal with the inputting flattened-dimension vectors. Without spatial information or CNN-based architectures, CAM-based tools are invalid to represent the internal learning pattern. To make such a concept versatile, we aim to propose another novel CAM-like tool with simple implementation that could be adopted in a wider range of backbones and problems. We note that the (softmax) outputs can replace the feature map $e(x)$ since the encoder $e$ could be directly regarded as $f$. Thus, it is fundamental to formulate an internal component for a value-based framework. For each candidate label, apart from its class output, its gradient of training loss with respect to the model output can also capture the importance degree of the candidate label to some extent. Inspired by the formulation of Grad-CAM in Eq. (2) that regards the gradient information flow to the last convolution layer as the importance of each neuron to the $j$-th class, we can intuitively derive the importance of each candidate label as follows:

$$v^j = \left| \frac{\partial(-\log(\psi^j(f(x))))}{\partial f^j(x)} \right| \psi^j(f(x)) = \left| \psi^j(f(x)) - 1 \right| \psi^j(f(x)), \tag{3}$$

where $\psi : \mathbb{R}^k \rightarrow \mathbb{R}^k$ is the activation function imposed on the logit vector $f(x)$ to approximate the class-conditional probability $P(y|x)$, and $\psi$ is mostly implemented by the *SoftMax* function. The

---

**Algorithm 1** CAVL Algorithm

---

**Input:** Model $f$, epoch $T_{\max}$, iteration $Z_{\max}$, partial labeled training set $\mathbb{D} = \{(\boldsymbol{x}_i, S_i)\}_{i=1}^m$.

    **for** $t = 1, 2, ..., T_{\max}$ **do**

        **Shuffle** $\mathbb{D}$;

        **for** $z = 1, 2, ..., I_{\max}$ **do**

            **Sample** mini-batch $\mathbb{D}_z = \{(\boldsymbol{x}_j, S_j)\}_{j=1}^u$ from $\mathbb{D}$;

            **if** $t = 1$ **then**

                **Update** $f$ by minimizing the cross entropy loss function $\hat{\mathcal{R}}_1$ in Eq. (6) with treating all the candidate labels from $S_j$ equally;

            **else**

                **Obtain** the CAV $v$ for each $\boldsymbol{x}_j$ by Eq. (4);

                **Generate** $y_j'$ for $\boldsymbol{x}_j$ by selecting the maximum $v$ from $S_j$ by Eq. (7);

                **Update** $f$ by minimizing the cross entropy loss function $\hat{\mathcal{R}}_{\text{cavl}}$ in Eq. (8) with treating $y_j'$ as the sole true label for each $\boldsymbol{x}_j$;

            **end if**

        **end for**

    **end for**

**Output:** $f$.

---

derivation process of the last inequality of Eq. (3) is provided in Appendix A. From Eq. (3), we can observe that the importance $v^j$ of each candidate label $j$ depends on the softmax confidence score $\psi^j(f(\boldsymbol{x}))$. However, we argue that such a calculation of $v^j$ is actually suboptimal, because $\psi^j(f(\boldsymbol{x}))$ conveys less information of model output than the logit $f^j(\boldsymbol{x})$ as $\psi^j(f(\boldsymbol{x}))$ could be considered as a condensed version of $f^j(\boldsymbol{x})$. Therefore, instead of directly adopting Eq. (3), we propose to replace $\psi^j(f(\boldsymbol{x}))$ in Eq. (3) by $f^j(\boldsymbol{x})$, and the importance of each candidate label would be calculated by the following simple way:

$$v^j = |f^j(\boldsymbol{x}) - 1|f^j(\boldsymbol{x}). \tag{4}$$

We call the importance $v^j$ calculated by Eq. (4) as the *class activation value* (CAV) of the $j$-th class, as it could be regarded as a value-based "CAM". Compared with CAM, we directly treat $f(\boldsymbol{x})$ as the "feature map" so that $v$ can be obtained by a weight imposed on the class output of the model. In summary, both CAV and CAM are the weighted operation to the feature maps (or outputs in the classifier). Differently from CAM, CAV is essentially a *value* so that it could be easily obtained and applied to different backbones. Besides, our CAV could also be generated with less time complexity since there is no need to process the weighted sum of the feature maps.

## 3.2 PARTIAL-LABEL LEARNING WITH CLASS ACTIVATION VALUE

In this section, we introduce how to effectively address PLL by using CAV in detail. Firstly, we give a formal introduction to PLL. For a $k$-class classification problem, let $\mathbb{X} \in \mathbb{R}^d$ be the feature (input) space and $\{\boldsymbol{x}_i\}_{i=1}^m \in \mathbb{X}$ be the input data, where $m$ refers to the number of training examples. Let $\mathbb{Y} = \{1, 2, ..., k\}$ be the label space. Let us denote $\mathbb{C} = \{2^{\mathbb{Y}} \backslash \emptyset \backslash \mathbb{Y}\}$ as the candidate label space where $2^{\mathbb{Y}}$ is the power set of $\mathbb{Y}$, and $|\mathbb{C}| = 2^k - 2$ shows that the candidate label set is not supposed to be the empty set nor the whole label set. For each training instance $\boldsymbol{x}_i$, let $y_i \in \mathbb{Y}$ be the ground truth label and $S_i \in \mathbb{C}$ be the candidate label set. We denote $P(\boldsymbol{x}, y)$ and $P(\boldsymbol{x}, S)$ as the probability densities of fully labeled examples and partially labeled examples respectively. Based on the crucial assumption of PLL that the candidate label set of each instance must include the correct label, we have $y_i \in S_i$, i.e.,

$$P(y_i \in S_i | y = y_i, \boldsymbol{x} = \boldsymbol{x}_i) = 1, \forall y_i \in \mathbb{Y}, \forall S_i \in \mathbb{C}. \tag{5}$$

PLL aims to learn a classifier $f : \mathbb{X} \rightarrow \mathbb{R}^k$ with training examples independently and identically sampled from $P(\boldsymbol{x}, S)$ to make correct predictions for test examples. To the best of our knowledge, most methods focus on designing the loss by considering all candidate labels or iteratively extracting the true label from the theoretical aspect. Differently from them, we concentrate on how to solve PLL by efficiently utilizing the learned representation by CAV, which could be potential and effective guidance for helping $f$ recognize the true label.

Note that the learned representation could be directly obtained by using CAV after firstly training $f$ in one epoch. For training $f$ in the first epoch, here we directly utilize the IM strategy, i.e., treating

every candidate label equally by cross entropy loss $\mathcal{L}(f(\boldsymbol{x}), s), s \in S$. Hence the empirical risk function $\hat{\mathcal{R}}_1$ for the first epoch could be defined as

$$\hat{\mathcal{R}}_1(\mathcal{L}, f) = \frac{1}{m} \sum_{i=1}^{m} \sum_{j \in S_i} \mathcal{L}(f(\boldsymbol{x}_i), s_j). \tag{6}$$

After training $f$ in one epoch, the CAV $v$ of every candidate label could be gained for each $\boldsymbol{x}$ by Eq. (4). To keep the classifier away from the negative effect by the false positive labels from the candidate set, we use the knowledge by CAV to guide the following learning process after the first training epoch, and the potential true label for $\boldsymbol{x}_i$ is selected by

$$y_i' = \arg\max_{j \in S_i} v_i^j, \tag{7}$$

where $y_i'$ is the selected true label from the candidate label set for $\boldsymbol{x}_i$. In this way, $y_i'$ could be straightly treated as the ground truth so that PLL could be transferred to classical supervised learning. Finally, the risk function $\hat{\mathcal{R}}_{\mathrm{cavl}}$ for the following epochs could be expressed as

$$\hat{\mathcal{R}}_{\mathrm{cavl}}(\mathcal{L}, f) = \frac{1}{m} \sum_{i=1}^{m} \mathcal{L}(f(\boldsymbol{x}_i), y_i'). \tag{8}$$

As $f$ gradually learns to approximate $P(y'|\boldsymbol{x})$, CAV would also be updated to identify the true label. Based on such a progressive label selection on the candidate label set, our Class Activation Value Learning (CAVL) is supposed to progressively training an accurate model with correctly selected labels. The pseudo-code of CAVL is given in Algorithm 1.

## 4 EXPERIMENTS

In this section, extensive experiments on various datasets are implemented to verify the effectiveness and rightness of our proposed CAV and CAVL method.

### 4.1 CAVL PERFORMANCE

#### 4.1.1 BENCHMARK DATASET COMPARISONS

**Datasets and backbones.** We use four popular benchmark datasets to test the performance of our CAVL, which are MNIST (LeCun et al., 1998), Fashion-MNIST (Xiao et al., 2017), Kuzushiji-MNIST (Clanuwat et al., 2018) and CIFAR-10 (Krizhevsky et al., 2009). Note that it is necessary to manually generate the candidate label sets since they are supposed to be used for single-classification problems. Recall that we introduce two different candidate label generation process (please refer to Section 2.2 for details) (Feng & An, 2019a;b; Feng et al., 2020; Lv et al., 2020; Wen et al., 2021), i.e., USS and FPS. The former one is implemented by uniformly sampling a label set from all the partial label space $\mathbb{C}$ for each instance, and the latter one sets a flipping probability $q$ to any irrelevant label so that it could become one item in the candidate label set with probability $q$. In this section we set $q \in \{0.3, 0.5, 0.7\}$ to represent different ambiguity degrees. For MNIST, Kuzushiji-MNIST and Fashion-MNIST, we adopt 3-layer MLP and 5-layer LeNet as the backbones. For CIFAR-10, we choose 34-layer ResNet (He et al., 2016) and 22-layer DenseNet (Huang et al., 2017) as the base models.

**Compared methods and training settings.** We compare our CAVL with four state-of-the-art approaches including PRODEN (Lv et al., 2020), Leveraged Weighted (LW) Loss (Wen et al., 2021), RC and CC (Feng et al., 2020). For all methods we search the initial learning rate from $\{0.0001, 0.001, 0.01, 0.05, 0.1, 0.5\}$, and weight decay from $\{10^{-6}, 10^{-5}, ..., 10^{-1}\}$. We take a mini-batch size of $\{32, 256\}$ images and train all the methods using Adam (Kingma & Ba, 2015) optimizer for 250 epochs. Hyper-parameters are searched to maximize the accuracy on a validation set containing 10% of the training samples annotated by true labels. All the implemented methods are trained on 1 RTX3090 GPU with 24 GB memory. To guarantee the fairness of comparisons, we repeatedly conduct all experiments 5 times and record the average accuracy with the standard deviation following the conventions in Feng et al. (2020); Lv et al. (2020); Wen et al. (2021). This work is partially supported by Huawei MindSpore (Huawei, 2020).

**Experiment results.** As shown in Table 1 and 2, our CAVL shows its superiority on four benchmark datasets with various backbones. Additionally, our CAVL outperforms nearly all state-of-the-art

Table 1: Test performance of CAVL and other methods on benchmark datasets using data generation by USS. The best results among all methods with the same backbone are marked in **bold**.

| Dataset (Backbones) | Method | Accuracy |
|---|---|---|
| MNIST (MLP/LeNet) | CC | $97.77 \pm 0.12\%$ / $98.77 \pm 0.06\%$ |
| | RC | $98.05 \pm 0.15\%$ / $99.03 \pm 0.04\%$ |
| | LW | $97.99 \pm 0.06\%$ / $98.74 \pm 0.05\%$ |
| | PRODEN | $97.02 \pm 0.08\%$ / $98.81 \pm 0.04\%$ |
| | CAVL | $\mathbf{98.06 \pm 0.05}\%$ / $\mathbf{99.14 \pm 0.03}\%$ |
| Kuzushiji-MNIST (MLP/LeNet) | CC | $88.87 \pm 0.32\%$ / $93.83 \pm 0.20\%$ |
| | RC | $\mathbf{89.36 \pm 0.30}\%$ / $\mathbf{94.01 \pm 0.15}\%$ |
| | LW | $87.98 \pm 0.39\%$ / $91.52 \pm 0.65\%$ |
| | PRODEN | $88.75 \pm 0.29\%$ / $93.94 \pm 0.18\%$ |
| | CAVL | $88.45 \pm 0.22\%$ / $93.25 \pm 0.21\%$ |
| Fashion-MNIST (MLP/LeNet) | CC | $87.90 \pm 0.27\%$ / $88.96 \pm 0.14\%$ |
| | RC | $88.40 \pm 0.13\%$ / $89.51 \pm 0.11\%$ |
| | LW | $88.16 \pm 0.12\%$ / $88.28 \pm 0.33\%$ |
| | PRODEN | $88.82 \pm 0.15\%$ / $89.23 \pm 0.12\%$ |
| | CAVL | $\mathbf{88.93 \pm 0.16}\%$ / $\mathbf{89.99 \pm 0.10}\%$ |
| CIFAR-10 (ResNet/DenseNet) | CC | $75.74 \pm 0.19\%$ / $76.78 \pm 0.33\%$ |
| | RC | $77.98 \pm 0.46\%$ / $78.56 \pm 0.37\%$ |
| | LW | $76.82 \pm 0.21\%$ / $78.08 \pm 0.66\%$ |
| | PRODEN | $77.62 \pm 0.34\%$ / $78.72 \pm 0.48\%$ |
| | CAVL | $\mathbf{78.05 \pm 0.32}\%$ / $\mathbf{79.10 \pm 0.25}\%$ |

Table 2: Test performance of CAVL and other methods on benchmark datasets using data generation by FPS. The best results among all methods with the same backbone are marked in **bold**.

| Dataset (Backbones) | Method | q=0.3 | q=0.5 | q=0.7 |
|---|---|---|---|---|
| MNIST (LeNet) | CC | $98.87 \pm 0.15\%$ | $98.49 \pm 0.07\%$ | $98.17 \pm 0.12\%$ |
| | RC | $98.88 \pm 0.07\%$ | $98.53 \pm 0.11\%$ | $98.12 \pm 0.05\%$ |
| | LW | $98.53 \pm 0.12\%$ | $98.68 \pm 0.06\%$ | $97.35 \pm 0.13\%$ |
| | PRODEN | $98.72 \pm 0.13\%$ | $98.62 \pm 0.09\%$ | $98.08 \pm 0.04\%$ |
| | CAVL | $\mathbf{98.90 \pm 0.12}\%$ | $\mathbf{98.71 \pm 0.04}\%$ | $\mathbf{98.20 \pm 0.11}\%$ |
| Kuzushiji-MNIST (LeNet) | CC | $93.11 \pm 0.08\%$ | $90.87 \pm 0.06\%$ | $89.98 \pm 0.14\%$ |
| | RC | $93.21 \pm 0.17\%$ | $91.19 \pm 0.22\%$ | $90.15. \pm 0.04\%$ |
| | LW | $92.65 \pm 0.18\%$ | $91.28 \pm 0.16\%$ | $\mathbf{90.55 \pm 0.17}\%$ |
| | PRODEN | $93.51 \pm 0.20\%$ | $91.23 \pm 0.17\%$ | $90.07 \pm 0.08\%$ |
| | CAVL | $\mathbf{93.82 \pm 0.21}\%$ | $\mathbf{91.57 \pm 0.11}\%$ | $86.05 \pm 0.25\%$ |
| Fashion-MNIST (LeNet) | CC | $89.41 \pm 0.17\%$ | $88.72 \pm 0.06\%$ | $85.87 \pm 0.25\%$ |
| | RC | $89.53 \pm 0.07\%$ | $88.84 \pm 0.14\%$ | $85.41 \pm 0.18\%$ |
| | LW | $89.19 \pm 0.08\%$ | $87.19 \pm 0.23\%$ | $85.92 \pm 0.13\%$ |
| | PRODEN | $89.63 \pm 0.05\%$ | $88.78 \pm 0.15\%$ | $85.81 \pm 0.16\%$ |
| | CAVL | $\mathbf{89.77 \pm 0.04}\%$ | $\mathbf{88.92 \pm 0.11}\%$ | $\mathbf{86.25 \pm 0.18}\%$ |
| CIFAR-10 (DenseNet) | CC | $77.32 \pm 0.14\%$ | $76.42 \pm 0.13\%$ | $66.17 \pm 0.25\%$ |
| | RC | $78.14 \pm 0.12\%$ | $77.42 \pm 0.16\%$ | $70.21 \pm 0.15\%$ |
| | LW | $80.95 \pm 0.17\%$ | $78.72 \pm 0.17\%$ | $\mathbf{71.26 \pm 0.16}\%$ |
| | PRODEN | $79.05 \pm 0.11\%$ | $77.52 \pm 0.18\%$ | $70.35 \pm 0.18\%$ |
| | CAVL | $\mathbf{81.58 \pm 0.22}\%$ | $\mathbf{79.69 \pm 0.17}\%$ | $65.86 \pm 0.21\%$ |

methods using the candidate labels generated by both USS and FPS, which shows that our CAVL does not rely on any data generation assumption. Specifically, it is worth mentioning that CAVL is able to show competitive performance with RC, which is theoretically proved to be possess optimal performance in data distribution generated by USS. Therefore, the results reasonably verify the generalization and effectiveness of our CAV and CAVL.

### 4.1.2 REAL-WORLD DATASET COMPARISONS

**Datasets and backbones.** We select five real-world datasets including Lost (Cour et al., 2011), MSRCv2 (Liu & Dietterich, 2012), BirdSong (Briggs et al., 2012), Soccer Player (Zeng et al., 2013) and Yahoo!News (Guillaumin et al., 2010). Note that these real-world datasets have their candidate labels. As a common practice in (Feng et al., 2020; Lv et al., 2020; Wen et al., 2021), we select

Table 3: Test performance of CAVL and other methods uisng linear model on real-world datasets. The best and second best results among all methods are marked in **bold** and underline.

| Method | Real-world datasets | | | | |
|--------|------|--------|----------|--------------|-----------|
| | Lost | MSRCv2 | Birdsong | SoccerPlayer | YahooNews |
| IPAL | 71.16 ± 2.56% | **50.64 ± 3.85%** | 70.32 ± 4.85% | 55.42 ± 0.92% | 66.43 ± 1.32% |
| CLPL | 75.01 ± 4.39% | 36.72 ± 4.61% | 64.35 ± 1.28% | 37.01 ± 1.02% | 45.21 ± 0.82% |
| PLSVM | 48.91± 3.33% | 35.95 ± 3.96% | 48.99 ± 1.98% | 45.90 ± 0.98% | 57.02 ± 1.02% |
| PLKNN | 37.73± 2.85% | 41.28 ± 2.25% | 64.32 ± 1.05% | 48.21 ± 1.21% | 40.67 ± 1.58% |
| PLECOC | 50.95 ± 7.81% | 43.25 ± 3.61% | 70.51 ± 4.31% | 55.29 ± 1.95% | 61.23 ± 1.52% |
| SURE | 75.56 ± 5.62% | 46.72 ± 4.21% | 55.42 ± 1.52% | 48.61 ± 0.84% | 55.17 ± 1.05% |
| LW | 79.74 ± 3.81% | 45.65 ± 3.12% | 72.01 ± 2.31% | 57.12 ± 3.25% | 67.94 ± 1.64% |
| RC | 80.45 ± 3.85% | 46.85 ± 2.52% | **72.15 ± 2.14%** | 57.05 ± 2.15% | 68.12 ± 0.67% |
| CC | 73.78 ± 4.34% | 45.13 ± 2.67% | 71.86 ± 1.34% | 56.54 ± 0.74% | 67.00± 0.35% |
| PRODEN | 80.12± 4.52% | 45.32 ± 2.38% | 71.90 ± 2.34% | 56.12 ± 3.12% | 67.92± 0.48% |
| CAVL | **82.07 ± 3.18%** | 48.39 ± 2.12% | 71.92 ± 0.73% | **57.32 ± 2.24%** | **68.89± 0.55%** |

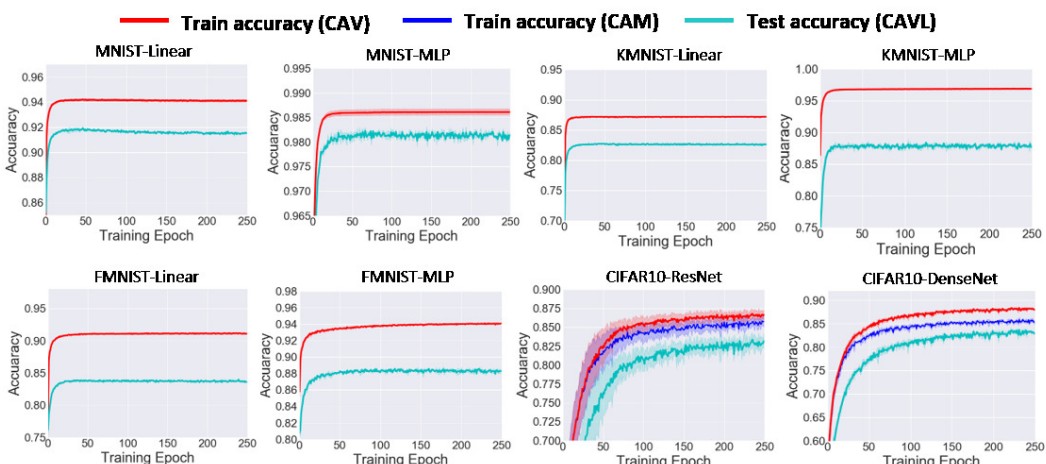

Figure 2: Study of CAV attributes on various backbones and datasets. Note that here CAM could only been obtained in deep CNN architectures, namely ResNet and DenseNet.

1-layer linear model and 3-layer MLP as the backbones to validate the performance of our CAVL in these five datasets.

**Compared methods and training settings.** Apart from the four methods mentioned in Section 4.1.1, we add SURE (Feng & An, 2019b), CLPL (Cour et al., 2011), IPAL (Zhang & Yu, 2015), PLSVM (Elkan & Noto, 2008), PLECOC (Zhang et al., 2017) and PLKNN (Hüllermeier & Beringer, 2006) as extra methods for comparison, where the parameters are searched to maximize the accuracy on a validation set (10% of the training set). We repeatedly run all experiments 10 times, and report the average accuracy with the standard deviation for each method. Other settings are similar to Section 4.1.1. The comparison results using MLP is recorded in Appendix B.2.

**Experiment results.** In Table 3, our proposed CAVL shows the best performance in nearly all five real datasets. Besides, CAM is unable to be extracted with the word embedding input such as BirdSong, lacking spatial information. Therefore, the achievement of CAVL on these real datasets shows the generalization and superiority of our CAVL to shallow model, where CAV could serve as the CAM to lead the classifier to learn the true label.

## 4.2 CAV ATTRIBUTES

In this part, we conduct several experiments to present how these attributes of CAV guide the learning process. We implement CAVL with four backbones including linear, MLP, ResNet and DenseNet on four benchmark datasets, whose candidate label sets are generated by USS. The training settings are similar to the ones in Section 4.1.1. Figure 2 illustrates the averaging performance of CAV, CAM (for deep networks) and the model itself during the 5-time training process. The red (blue) line in Figure 2 depicts the training accuracy measured by treating the maximum CAV (CAM) from the

candidate set as the true label, and the cyan line depicts the test accuracy of the model. We note that here CAM is simply extracted from the classifier for comparison, which is different from its usage in Section 2.2. As shown in Figure 2, it is clear that the classifier learns to approximate the better data distribution by CAV, and the accuracy trend recognized by CAV is consistent with the classifier performance in the test datasets, which validates the `Power` and `Dynamic` attribute of CAV.

## 5 RELATED WORK

In this section, we give a brief introduction to the two mainstream strategies for *partial-label learning* (PLL), i.e., the *averaged-based strategy* (ABS) and the *identification-based strategy* (IBS).

ABS treats all candidate labels in an equal manner and then averages the model outputs of all candidate labels for prediction. Some non-parametric methods (Hüllermeier & Beringer, 2006; Gong et al., 2017) concentrated on predicting the label by leveraging the outputs of its neighbors. Furthermore, some methods (Cour et al., 2009; Zhang et al., 2016; Yao et al., 2020) aimed to subtly leverage the labels outside the candidate set so as to discriminate the potential true label. Yao et al. (2020) designed an entropy-based regularizer to minimize the entropy of each label to maximize the margin between the potential true label and the unlikely labels. A recent study (Feng et al., 2020; Lv et al., 2020; Wen et al., 2021; Lv et al., 2021) focused on the data generation process and proposed a classifier-consistent method based on a transition matrix. (Chen et al., 2020) proposed a new problem General Partial Label Learning and leveraged graph neural network to tackle the label ambiguity.

IBS aims to constantly identify the most possible true label from the candidate label set to eliminate the label ambiguity. Early works treated the potential truth label as a latent variable and optimized the objective function by the maximum likelihood criterion (Jin & Ghahramani, 2002; Liu & Dietterich, 2014) or the maximum margin criterion (Nguyen & Caruana, 2008; Yu & Zhang, 2016). Later, taking advantage of topological information of feature space to generate the score of each candidate label (Zhang & Yu, 2015; Zhang et al., 2016; Feng & An, 2018; Wang et al., 2019) attracted much attention from researchers. These methods commonly aimed to iteratively update the confidence of each candidate label based on the widely used assumption that similar instance are supposed to possess the same label. Xu et al. (2019b) proposed to model the generalized label distribution by leveraging the topological information of the feature space. Lyu et al. (2021) reformulated PLL into a matching selection problem and proposed a Graph-Matching based Partial Label Learning framework to solve the problem. It is worth noting that IBS is commonly susceptible to the false positive labels that co-occur with the true label in the candidate label set. Thus our CAVL, which belongs to IBS, also owns this inevitable shortcoming.

## 6 CONCLUSION

In this paper, we exploited the learned intrinsic representation of the model for *partial-label learning* (PLL). We made two key contributions. Firstly, we found that the *class activation map* (CAM), a simple technique for discriminating the learning patterns of each class in an image, could surprisingly be used to differentiate the true label from candidate labels. Unfortunately, we are yet unable to directly use CAM for PLL, as CAM is subject to image inputs with convolutional neural networks. Thus, our second contribution is to propose the *class activation value* (CAV), which not only owns similar properties of CAM but also is versatile in various types of inputs and models. Based on CAV, we proposed a simple yet effective PLL method that selects the true label by the class with the maximum CAV for model training. Comprehensive experimental results on various datasets demonstrated that our CAVL achieved state-of-the-art performance.

ACKNOWLEDGMENT

FZ and BH were supported by NSFC Young Scientists Fund No. 62006202, RGC Early Career Scheme No. 22200720 and HKBU CSD Departmental Incentive Grant. BH was also supported by MSRA StarTrack Program. LF was supported by the National Natural Science Foundation of China (Grant No. 62106028) and CAAI-Huawei MindSpore Open Fund. TLL was supported by Australian Research Council Projects DE-190101473 and DP-220102121. MS was supported by JST CREST Grant Number JPMJCR18A2.

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

## A    DERIVATION PROCESS OF EQUATION. (3)

The *SoftMax* function is $\psi^j(x) = \exp(x_j)/\sum_{j=1}^{k}\exp(x_j)$, where $x \in \mathbb{R}^d$ refers to the. Then we can obtain the derivation of $-\log(\psi^j(x))$ with respect to the $x_j$ by

$$
\begin{aligned}
\frac{\partial(-\log(\psi^j(x)))}{\partial x_j} =& \frac{\partial(-\log(\psi^j(x)))}{\partial \psi^j(x)}\frac{\partial \psi^j(x)}{\partial x_j} \\
=& -\frac{1}{\psi^j(x)}\frac{\partial \psi^j(x)}{\partial x_j} \\
=& -\frac{1}{\psi^j(x)}\frac{\sum_{i=1}^{d}e^{x_j}e^{x_i} - (e^{x_i})^2}{\sum_{i=1}^{d}(e^{x_i})^2} \\
=& -\frac{1}{\psi^j(x)}\left(\frac{e^{x_j}}{\sum_{i=1}^{d}(e^{x_i})^2}\right)\left(1 - \frac{e^{x_j}}{\sum_{i=1}^{d}(e^{x_i})^2}\right) \\
=& -\frac{1}{\psi^j(x)}\psi^j(x)(1 - \psi^j(x)) \\
=& \ \psi^j(x) - 1.
\end{aligned}
\tag{9}
$$

## B    DETAILED SUPPLEMENTARY FOR EXPERIMENTS

### B.1    BENCHMARK DATASETS

In Section 4.1.1, we use four widely-used benchmark datasets, i.e. MNIST (LeCun et al., 1998), Fashion-MNIST (Xiao et al., 2017), Kuzushiji-MNIST (Clanuwat et al., 2018) and CIFAR-10 (Krizhevsky et al., 2009). Table 4 lists the characteristics of these datasets. We respectively describe these datasets as follows.

- MNIST: It is a 10-class dataset of handwritten digits. Each data is a 28 × 28 grayscale image.
- Fashion-MNIST: It is also a 10-class dataset. Each instance is a fashion item from one of the 10 classes, which are T-shirt/top, trouser, pullover, dress, sandal, coat, shirt, sneaker, bag, and ankle boot. Moreover, each image is a 28 × 28 grayscale image.
- Kuzushiji-MNIST: Each instance is a 28 × 28 grayscale image associated with one label of 10-class cursive Japanese ("Kuzushiji") characters.
- CIFAR-10: Each instance is a 32 × 32 × 3 colored image in RGB format. It is a ten-class dataset of objects including airplane, bird, automobile, cat, deer, frog, dog, horse, ship, and truck.

### B.2    REAL DATASETS

In Section 4.1.2, We select five real-world datasets including Lost (Cour et al., 2011), MSRCv2 (Liu & Dietterich, 2012), BirdSong (Briggs et al., 2012), Soccer Player (Zeng et al., 2013) and Yahoo!News (Guillaumin et al., 2010). Here we make a comprehensive descriptions about them shown as follows.

- Lost, Soccer Player and Yahoo! News: They crop faces in images or video frames as instances, and the names appearing on the corresponding captions or subtitles are considered as candidate labels.

- MSRCv2: Each image segment is treated as a sample, and objects appearing in the same image are regarded as candidate labels.

- BirdSong: The singing syllables of birds are regarded as instances and bird species who are jointly singing during any ten seconds are represented as candidate labels

Table 4: Characteristics of benchmark datasets

| Datasets | #Train | #Test | #Features | #Classes |
|---|---|---|---|---|
| MNIST | 60,000 | 10,000 | 784 | 10 |
| Fashion-MNIST | 60,000 | 10,000 | 784 | 10 |
| Kuzushiji-MNIST | 60,000 | 10,000 | 784 | 10 |
| CIFAR-10 | 50,000 | 10,000 | 3072 | 10 |

Table 5: Characteristics of real-world datasets

| Datasets | Application Domain | #Examples | #Features | #Classes | Avg #CLs |
|---|---|---|---|---|---|
| Lost | Automatic face naming | 1,122 | 108 | 16 | 2.23 |
| MSRCv2 | Object classification | 1,758 | 48 | 23 | 3.16 |
| BirdSong | Bird song classification | 4,998 | 38 | 13 | 2.18 |
| Soccer Player | Automatic face naming | 17,472 | 279 | 171 | 2.09 |
| Yahoo! News | Automatic face naming | 22,991 | 163 | 219 | 1.91 |

Table 6: Test performance of the CAVL and other methods uisng MLP on real-world datasets. The best and second best results among all methods are marked in **bold** and underline.

| Method | Real-world datasets | | | | |
|---|---|---|---|---|---|
| | Lost | MSRCv2 | Birdsong | SoccerPlayer | YahooNews |
| LW | $67.53 \pm 3.21\%$ | $50.54 \pm 3.23\%$ | **$71.74 \pm 1.35\%$** | $53.29 \pm 1.95\%$ | $65.94 \pm 0.92\%$ |
| RC | **$75.89 \pm 5.10\%$** | $\underline{51.13 \pm 2.67\%}$ | $70.14 \pm 1.54\%$ | $53.98 \pm 0.85\%$ | $\underline{67.56 \pm 2.53\%}$ |
| CC | $70.10 \pm 5.20\%$ | $50.57 \pm 3.47\%$ | $\underline{70.51 \pm 1.41\%}$ | **$54.87 \pm 0.25\%$** | **$67.75 \pm 1.12\%$** |
| CAVL | $\underline{71.82 \pm 3.24\%}$ | **$51.55 \pm 3.31\%$** | $69.25 \pm 1.44\%$ | $\underline{54.83 \pm 0.74\%}$ | $65.32 \pm 1.04\%$ |

Table 5 shows the average number of candidate labels (Avg. # CLs) per instance. Table 6 presents the performance comparison between other methods and our CAVL method based on MLP model, the results of which also validate the effectiveness our CAVL.

## B.3 ABLATION STUDIES ON THE STARTING EPOCH IN CAVL

In Section 3.2, we propose the CAVL method to address PLL. Specifically, we use the IM to train the classifier in the first epoch, and then the potential true labels could be obtained with CAV after the following training epochs. Here we set the epoch number for starting using the CAVL method, i.e. the starting epoch, as one. To further validate the effectiveness of our CAVL, we explore the effects of the starting epoch on our method. Specifically, we implement CAVL with two backbones including LeNet and ResNet on four benchmark datasets. We repeatedly run the experiments 5 times by using candidate labels generated by USS. Following similar training settings in Section 4.1.1, we set the total training epoch as 250, and select the starting epoch from $[10, 50, 100, 150]$. IM is used before the selected starting epoch and CAVL is implemented after the selected epoch.

As shown in Table 7, the accuracy of the classifier becomes lower as the increase of the starting epoch in CAVL. Let us take the results in KMINIST as an example, the CAVL shows its superiority with setting the starting epoch as 1, achieving $93.25 \pm 0.21\%$ accuracy performance. As the starting epoch increases, the classifier is more negatively affected by the false positive labels in the candidate sets and dragged away from the true labels by IM. Thus it only achieves $88.54 \pm 0.26\%$ with the starting epoch as 150. This phenomenon is reasonable because the DNN learn patterns first, which suggests that deep networks can gradually memorize the data, moving from regular data to irregular data such as outliers and mislabeled data (Arpit et al., 2017). Therefore, our CAVL could not

Table 7: The performance of our CAVL with different settings of the starting epoch.

| Starting Epoch | Dataset (Backbones) | | | |
|:---:|:---:|:---:|:---:|:---:|
| | MNIST (LeNet) | KMNIST (LeNet) | FMNIST (LeNet) | CIFAR-10 (ResNet) |
| 1 | **99.14 ± 0.03%** | **93.25 ± 0.21%** | **89.99 ± 0.10%** | **78.05 ± 0.32%** |
| 10 | 97.82 ± 0.04% | 92.84 ± 0.21% | 88.75 ± 0.14% | 75.27 ± 0.42% |
| 50 | 97.25 ± 0.03% | 91.23 ± 0.25% | 87.98 ± 0.15% | 72.17 ± 0.31% |
| 100 | 96.44 ± 0.07% | 89.66 ± 0.42% | 86.92 ± 0.16% | 71.55 ± 0.18% |
| 150 | 96.38 ± 0.07% | 88.54 ± 0.26% | 86.07 ± 0.34% | 69.87 ± 0.12% |

sufficiently help the classifier to deal with this memorization effect during the rest of the training epochs. In conclusion, the effectiveness of our CAVL would be weakened as extending the starting epoch.

### B.4 GENERATION OF CANDIDATE LABELS

In Section 4.1.1 we introduce two different generation ways for the candidate label sets, i.e, USS, Uniformly sampling a label set from all the partial label space $\mathbb{C}$ for each instance. FPS, Setting a flipping probability $q$ to any irrelevant label which could possibly become one item in the candidate label set with probability $q$.

For USS, each partially labeled example $(\boldsymbol{x}, S)$ is independently drawn from a probability distribution with the following density:

$$\widetilde{P}(\boldsymbol{x}, S) = \sum_{i=1}^{k} P(S|y=i)P(\boldsymbol{x}, y=i), P(S|y=i) = \begin{cases} \frac{1}{2^{k-1}-1} & i \in S, \\ 0 & i \notin S. \end{cases} \qquad (10)$$

The generation process assumes that the candidate label set $S$ is independent of the instance $\boldsymbol{x}$. There are totally $2^k - 1$ possible candidate label sets that contain the specific true label $y$. Therefore, Eq. (10) illustrates the candidate label set for each instance is uniformly sampled.

For FPS, we set a flipping probability $q$ to any irrelevant label that possibly entries the candidate label set. Here we introduce the class transition matrix (denoted by $T$) for partially labeled data, where $T_{ij}$ refers to the probability of the label $j$ being a candidate label given the true label $i$ for each instance. Note that $T_{ii} = 1$ always holds since the true label always belongs to the candidate label. $T_{ij} = q, i \neq j$ holds for other elements. The matrix representation of $T$ is expressed as:

$$\begin{bmatrix} 1 & q & q & q & q & q & q & q & q & q \\ q & 1 & q & q & q & q & q & q & q & q \\ q & q & 1 & q & q & q & q & q & q & q \\ q & q & q & 1 & q & q & q & q & q & q \\ q & q & q & q & 1 & q & q & q & q & q \\ q & q & q & q & q & 1 & q & q & q & q \\ q & q & q & q & q & q & 1 & q & q & q \\ q & q & q & q & q & q & q & 1 & q & q \\ q & q & q & q & q & q & q & q & 1 & q \\ q & q & q & q & q & q & q & q & q & 1 \end{bmatrix}$$

### B.5 COMPARED METHODS

The compared PLL methods are listed as follows.

- SURE (Feng & An, 2019b): It iteratively enlarges the confidence of the candidate label with the highest probability to be the correct label.

- CLPL (Cour et al., 2011): It uses a convex formulation by using the one-versus-all strategy in the multi class loss function.

- IPAL (Zhang & Yu, 2015): It is a non-parametric method that applies the label propagation strategy to iteratively update the confidence of each candidate label.
- PLSVM (Elkan & Noto, 2008): It is a maximum margin-based method that differentiates candidate labels from non-candidate labels by maximizing the margin between them.
- PLECOC (Zhang et al., 2017): It adapts the Error-Correcting Output Codes method to deal with partially labeled examples in a disambiguation-free manner.
- PLKNN (Hüllermeier & Beringer, 2006): It adapts the widely-used *k-nearest neighbors* method to make predictions for partially labeled examples.
- CC & RC (Feng et al., 2020): The former method is a novel risk-consistent partial label learning method and the latter one is classifier-consistent based on the generation model.
- LW (Wen et al., 2021): The method proposes a family of loss function for the first time, where introduces the leverage parameter $\beta$ to consider the trade-off between losses on partial labels and non-partial labels.

For all the above methods, their parameters are specified or searched according to the suggested parameter settings by respective papers to maximize the accuracy performance on a validation set, which is made by the 10% of the training set.

### B.6 DETAILS OF THE BACKBONES

In Section 4 we select five network architectures as the backbones for modelling our classifier, which are linear, 3-layer MLP, 5-layer LeNet, 34-layer ResNet (He et al., 2016) and 22-layer DenseNet (Huang et al., 2017). The linear model is a linear-in-input model:$d - k$. MLP refers to a 3-layer fully connected networks with ReLU as the activation function, and the architecture is $d - 500 - k$. LeNet is comprised of 2 Convolution Layer and 3 fully connected layer. 34-layer ResNet and 22-layer DenseNet strictly follow the implementation by He et al. (2016) and Huang et al. (2017).

### B.7 PILOT EXPERIMENTS IN REAL-WORLD DATATSETS

In Section 2.2 we present the pilot experiments on CIFAR10, which shows that CAM possess two attributes, i.e., `power` and `dynamic`. Here we provide more results on five real-world datasets to prove the rightness of our proposed CAV. The real-world datasets already have their candidate labels. Following similar training settings in Section 2.2, we select 1-layer linear as the backbone for the classifier and respectively use the IM and CAV to train it in 10 epochs. Intuitively, using CAV to train the model is equal to CAVL. Note that CAM could not be obtained in the linear module. Figure 3 shows the average results of 10-time trials and it is easily found that CAV also shows `power` and `dynamic` attributes in real-world datasets, validating the generalization and effectiveness of our CAV.

## C FURTHER EXPLANATIONS ON CLASS ACTIVATION MAP

Here we provide a detailed explanation of CAM. For a well-trained CNN module for addressing the classification problem, the Class Activation Map (CAM) is simply a weighted linear sum of the presence of the visual patterns at different spatial locations for the input images. CAM aims to mine out what the network focuses on the input images related to true labels during the training phase. It is known that a CNN-based module is a "black box" and the meaning behind CAM is the exploration of the inner principles inside such technology. Based on CAM, it is convenient to visualize the learning patterns of the CNN-based modules. Figure 4 shows several CAM results selected from (Zhou et al., 2016). Here we take one example of the Teapot for illustration, the CAM of the image related to the teapot shows the most discriminative regions in the head part, which manifests that the classification classifies the image to the teapot since it finds and concentrates on the head part of the teapot. Thanks to CAM, we could conveniently and effectively investigate the inner principles of CNN-based modules by simply utilizing the components from the network itself. This is also the reason why we name it "internal representation", which is essentially a form of intermediate outputs.

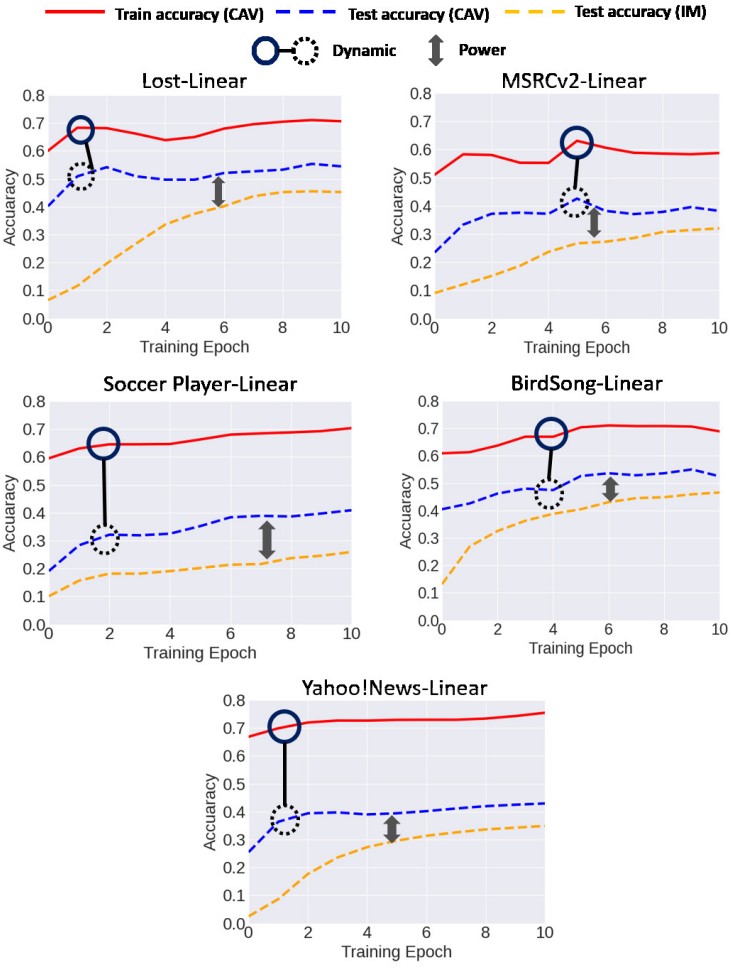

Figure 3: Comparison of accuracy performance on five real-world datatsets obtained by IM and CAVL. Note that the training method is IM.

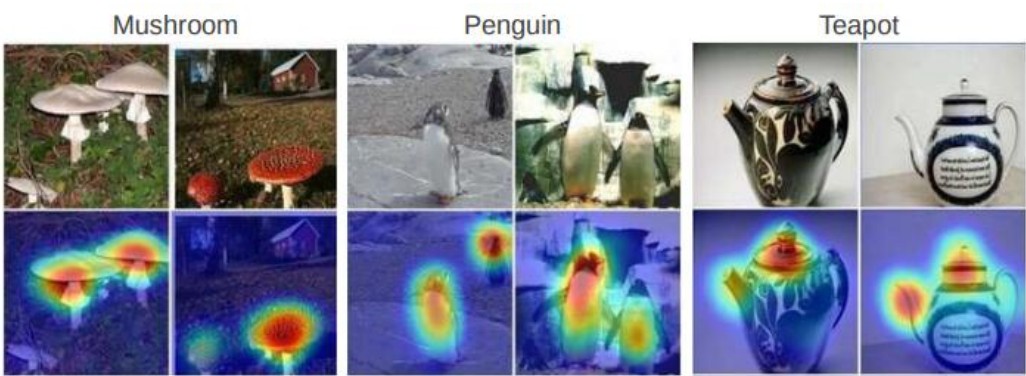

Figure 4: Some CAM samples selected from (Zhou et al., 2016).

