# OpenReview forum: "Exploiting Class Activation Value for Partial-Label Learning"
_ICLR.cc/2022/Conference — ICLR 2022 Poster_

### Official Review · Reviewer_sUUe · 2021-10-27

**Correctness:** 2
**Technical Novelty And Significance:** 2
**Empirical Novelty And Significance:** 2
**Recommendation:** 3
**Confidence:** 4

**Main Review:**

Strengths:
1.	This paper introduces the class activation map (CAM) for handling identification in PLL.
2.	This paper proposes the class activation value (CAV) for identification via capturing the learned representation information in a more general way instead of CAM as CAM is confined for image datasets and CNN model.

Weakness:
1.	The authors claim that “We believe that such a mechanism (Grad-CAM or CAM) could guide f to differentiate the true label from the candidate set because it is constructed by taking advantage of internal elements in f” and conducted a pilot experiment to check whether CAM with the most foreground seeds belongs to the true label. However, the pilot experiment cannot empirically validate this claim, as the experiment is only conducted on the CIFAR-10 with artificial candidate labels generated by two simple strategies but not conducted on real-world PLL datasets or adopts some realistic candidate labels generation strategies such as class-dependent strategy.
2.	The proposed method CAVL adopts the strategy that selects the “true” label from the candidate label set and only considers the loss on the “true” label after the first training epoch. However, the selected “true” label may be false-positive and the authors also claim that the potential true label would be updated. In this case, the training model could be affected by false-positive labels since CAVL only considers the “true” label.  [1] proves that models trained by this strategy cannot achieve good performance.
3.     There is no theoretical result provided to validate the effectiveness of CAV for PLL.
4.	The authors adopt the PLL experimental settings in [1], but they do not compare the method in [1].
5.	The authors claim that the hyper-parameters are selected to exploit the best performance on the validation set containing 10% of the training examples. They should show the train/ validation/ test split. RC, CC and CAVL are suggested to have the same learning rate, weight decay and mini-batch size as they adopt the same model for fair comparisons.


Suggestion:
The authors are suggested to provide source code as supplementary material so that the reviewers could check the reproducibility of the experimental results.

[1] Progressive Identification of True Labels for Partial-Label Learning, ICML 2020.

**Summary Of The Paper:**

This paper focuses on the problem of partial-label learning (PLL), where each training instance is assigned a set of candidate labels that include the true label.  This paper shows that class activation map (CAM) could identify the true label from candidate labels for PLL. In addition, the authors propose the class activation value (CAV) for identification via capturing the learned representation information in a more general way instead of CAM as CAM is confined for image datasets and CNN model. CAV Learning that selects the true label by the class with the maximum CAV for model training is proposed. Experiments on various datasets demonstrate that the proposed method achieves state-of-the-art performance.

**Summary Of The Review:**

This paper introduces the class activation map (CAM) for handling identification in PLL. But the effectiveness of CAM for PLL is not validated. Several of the paper’s claims are incorrect or not well-supported. The contributions are only marginally novel.

---

> ### Author Response · Authors · 2021-11-14
> **Response to Reviewer sUUe (R3) (PART 1/2)**
>
> Thank you for your detailed suggestions. Below are our responses to your comments/questions.
> ***
> **Q1: Lack of real-world datasets in pilot experiments.**
>
> **A1**: Thanks for your constructive suggestions! We have added more pilot experiments based on
> real-world datasets in ***Appendix B.7***. The results also show that our CAV possesses
> the dynamic and power attributes on real-world datasets. ***Figure 1*** simply serves as an
> illustration for our CAV and CAM. Due to the page limitation, we did not provide
> such illustrations on all datasets. Here we respectfully remind the reviewer that
> our CAVL is demonstrated to be effective by sufficient experiments in ***Section 4***, which is also
> acknowledged by ***R1*** and ***R2***. The ablation studies on CAV in ***Figure 2*** also
> show that CAV and CAM could possess the power and dynamic attributes among all benchmark datasets.
> Thus we believe that the generalization and rightness of the two attributes of our CAV have been well verified.
> ***
> **Q2: Influence by false positive labels.**
>
> **A2:** Thank you for raising this concern. We agree with you that our CAVL could
> possibly select false positive labels. However, we would like to emphasize that
> ***all IBS-based methods could not avoid the negative effects by the false positive labels***,
> which is a common challenge in PLL [4-7]. Our CAV could well relieve this issue due to
> its two observed attributes, namely power and dynamic. According to the pilot experiments in ***Figure 2***
> and ***Appendix B.7***, our CAV is obviously capable of selecting more true labels than the
> classifier itself. Besides, the experimental results in ***Section 4*** also sufficiently validates
> the effectiveness and superiority of our CAVL. Thus we believe our CAVL could
> ***effectively*** deal with the negative influence from the false positive labels.
>
> We ***partially disagree with your comment made from [10]***. Lv *et al*. [10] proposed PRODEN,
> which essentially is the weighted cross entropy loss. For the selection of weights, Lv *et al*. [10]
> reasonably considered the contribution by all the labels from the candidate set. Besides, Lv *et al*. [10]
> designed another version of PRODEN for comparison, namely the PRODEN-sudden. PRODEN-sudden treated
> the label with the maximum output of the classifier as the true label, and set its corresponding
> weight as 1 (the rest of the weights were 0). Lv *et al*. [10] had a clear description for
> it in Section 5 that "...PRODEN-sudden means performing a sudden identification, i.e.,
> updating the weights $w_{ij} = 1$ if $argmax_{j \in s_{i}} \ g_{j}(x_{i}) = k$ and $w_{ij} = 0,
> \forall j \neq k$ in every iteration step...". The experimental results indeed
> showed that PRODEN-sudden achieves worse performance than PRODEN. However, we would like to
> respectively remind the reviewer that you might overlook that
> ***the PRODEN-sudden selected the label with the maximum normal output of the classifier as the true label***.
> According to our pilot experiments in ***Section 2*** and ***Appendix B.7***,
> it obviously showed that the classifier itself was easily affected by the false negative labels
> but CAV and CAM could recognize more accurate labels than the normal outputs due to their power attributes.
> Thus we also contend that the classifier could not learn the true label distribution by directly relying
> on the model outputs by itself, and this is also the reason why we treated the label judged
> by CAV as the true labels. In fact, the conclusion in [9] ***indirectly supports our
> effectiveness of CAV and CAVL*** since we did not regard the maximum output of the classifier as the true label.

---

> > ### Author Response · Authors · 2021-11-14
> > **Response to Reviewer sUUe (R3) (PART 2/2)**
> >
> > **Q3: Lack of theoretical analysis.**
> >
> > **A3**: Thank you for this comment and we admit that our CAVL is not theoretically grounded.
> > However, we would argue that our CAVL is well supported by ***empirical findings*** and
> > such empirical findings are revealed in our paper ***for the first time***. This research pattern
> > could also be found in many frameworks in machine learning and deep learning (which also fits the scope of
> > ICLR). For instance, CAM is purely the weighted sum of feature maps without any theoretical support [8] but has
> > been successfully applied in numerous tasks in Computer Vision for visualizing the internal
> > knowledge of CNN-based networks [11-13]. Besides, some methods in PLL also are proposed
> > without theoretical support. Yao *et al*. [14] proposed to utilize
> > two networks to interactively select the true labels from complicated
> > and simple groups. Yao *et al*. [5] found that the “...margin between
> > the largest label output and the second largest label output in label
> > vector increased during the training phase...", indicating that the
> > uncertainty of the label vector decreased. In this way, an entropy-based
> > regularizer was designed to maximize such margin to highlight the potential true label.
> > Yao *et al*. [5] also mentioned that “...all existing approaches are non-deep,
> > which means that they only work on the handcrafted features and the performances are
> > far from perfect in many cases...”, which illustrates another limitation to those methods
> > based on strict theoretical analysis. According to the experimental results, our CAVL could
> > adapt to ***any backbones without relying much on the label generation process***. Therefore,
> > we believe our CAVL is meaningful to address the PLL despite lacking theoretical analysis.
> > We also sincerely agree that ***theoretical support is important***, and we will try to
> > provide theoretical guarantees in our future work.
> > ***
> > **Q4: Lack of compared method.**
> >
> > **A4**: Thank you sincerely for this suggestion! We have added additional
> > experiments to make comparison with this method in ***Section 4***.
> > ***
> > **Q5: Settings of experiments.**
> >
> > **A5**: Thank you for raising this concern. For the splitting of the real-world datasets,
> > we strictly follow the settings from [10,15] for a fair comparison.
> > We split the train/test as 90\%/10\% for the five real-world datasets.
> > For all datasets including four benchmark datasets and five real-world datasets,
> > the hyper-parameters are searched to maximize the accuracy on a validation set containing 10\% of the
> > partially labeled training samples.
> >
> > For the hyper-parameter configurations of all methods, we respectfully ***disagree with the
> > reviewer that RC, CC, and CAVL should have the same hyper-parameters for training***. Firstly,
> > many approaches in PLL used hyper-parameters that maximize the performance of the compared
> > methods [7,10,16,17]. Yan *et al*. [7] and Lv *et al*. [10] all stated in the experiment
> > part that “...their hyper-parameters are selected according to the suggested
> > settings in original papers...”. Wang *et al*. [16] stated in Section 4.1 that “...each
> > configured with parameters suggested in respective literatures...”. Zhang *et al*. [17]
> > stated in Section 4.1 that "... Each compared algorithm is implemented with the default
> > hyper-parameter setups suggested in respective literatures...". Secondly, it may cause
> > ***unfair comparison*** if the same hyper-parameters are used. For example,
> > following the same hyper-parameters in CAVL, RC with MLP model could achieve only
> > $97.65 \pm  0.07\%$ accuracy in MINIST with USS label generation, and the
> > result is lower than using the parameters suggested in the paper. The readers
> > would reasonably doubt that the hyper-parameters could be deliberately selected to
> > maximize the performance of CAVL by lowering the performance of other methods.
> > Thus, we have implemented the ground search for the hyper-parameters.
> > In conclusion, we believe that such an experimental setting would be ***fair***.

---

> > > ### Comment · Reviewer_sUUe · 2021-11-19
> > > **Discussion**
> > >
> > > 1. The authors claim that "the hyper-parameters are searched to maximize the accuracy on a validation set containing 10% of the partially labeled training samples".  The task of the PLL model is predicting the true label but the partially labeled samples (the true label is unidentified) cannot offer the validation for searching the hyper-parameters of the model. The claim is self-contradictory.
> > >
> > > 2. The only contribution of this paper is CAV, which is the minor modification of CAM [1]. The novelty is limited and the paper only gives the empirical findings on CAV. Why CAV $|\phi^j(f^j(x))-1|\phi^j(f^j(x))$ can identify more true labels than $\phi^j(f^j(x))$ [2]?  Much more analysis and empirical results should be added.
> > >
> > > [1] Learning deep features for discriminative localization, CVPR 2016.
> > >
> > > [2] Progressive Identification of True Labels for Partial-Label Learning, ICML 2020.

---

> > > > ### Author Response · Authors · 2021-11-19
> > > > **Response to Reviewer sUUe (R3)**
> > > >
> > > > **Q1**. Claim about the validation set.
> > > >
> > > > **A1**. Thanks for your comment. We would like to clarify that our claim is ***not "self-contradictory"***.
> > > > In fact, it is simply ***common sense*** that the accuracy could not be obtained without true labels.
> > > > For searching the hyper-parameters that maximize the accuracy on the validation set,
> > > > we ***certainly*** utilized the true labels in the candidate labels for calculating the accuracy.
> > > > Besides, this claim could also be found in related work. Wen *et al*. [4] clearly
> > > > stated in Section 4.4.1 that “...hyper-parameters are searched to maximize the accuracy
> > > > on a validation set containing 10\% of the partially labeled training samples...”, which is
> > > >  ***exactly the same as our claim***. It is also worth noting that we never used
> > > >  true labels in the model training process, and the true labels were only used
> > > >  for performance evaluation. Thus, we reasonably ***do not accept this groundless questioning***.
> > > > ***
> > > >
> > > > **Q2**. Limited contribution and novelty.
> > > >
> > > > **A2**. Thanks for your comment. We ***firmly disagree with your comment*** that
> > > > "The only contribution of this paper is CAV". In fact, we have repeatedly emphasized that
> > > > our ***first contribution*** is the finding of the positive effect of CAM on addressing PLL,
> > > > which is acknowledged by both ***R1*** and ***R2***. Recognizing that CAM is not versatile in wide applications,
> > > > we further proposed CAV for the substitution and the qualitative and quantitative experiments
> > > > already validated the effectiveness of our CAV and CAVL, which is also agreed by ***R1*** and ***R2***.
> > > > Although we admit that our model lacks theoretical analysis, we have ***clearly explained our
> > > > motivation*** for CAM and CAV by supportive empirical results, as shown in Figure 1.
> > > > In fact, here we would like to reiterate that we ***sincerely agree that it is important to find theoretical
> > > > guarantees***, and we will insist on exploring it in the future.

---

> > > > > ### Comment · Reviewer_sUUe · 2021-11-20
> > > > > **Response**
> > > > >
> > > > > 1. You claim that this paper utilized  "partially labeled training samples"  for validation, i.e.,  the samples are annotated by the candidate labels and the true labels are unidentified. How can you search the hyper-parameters without true labels? I think the claim is incorrect  and should be changed as "the hyper-parameters are searched to maximize the accuracy on a validation set containing 10% of the training samples annotated by true labels."
> > > > > 2.  CAV is the minor modification of CAM.  Why CAM  with minor modification can work for PLL? Why CAV $|\phi^j(f^j(x))-1|\phi^j(f^j(x))$ can identify more true labels than common technique $\phi^j(f^j(x))$? The empirical result can only be one of the supportive materials but cannot serve as an analysis. If these core questions are not answered, the paper just utilizes the technique CAM  in CV and modifies it for PLL with the empirical results, where the novelty is limited.

---

> > > > > > ### Author Response · Authors · 2021-11-20
> > > > > > **Response to Reviewer sUUe (R3)**
> > > > > >
> > > > > > Q1. Claim about the validation set.
> > > > > >
> > > > > > A1. Thanks for this comment and we are glad that we ***reach an agreement*** on the "settings of the validation". We have modified the relevant claims in ***Section 4*** as your suggestion, to make it clearer.
> > > > > >
> > > > > > Q2. Support of our CAV and CAVL.
> > > > > > ***
> > > > > > A2. Thanks for this comment. In fact, we contend that the empirical results in ***Figure 1*** and ***Appendix B.7*** could provide the motivation/explanation why our CAM and CAV are better than the normal outputs. The reason why we tried to use CAM as the guidance to address PLL is driven by the application of CAM in Computer Vision, which mostly represents the ***learning patterns of the deep networks***. The meaning behind CAM is the exploration of such "black box" technology. This is also why CAM is widely used in Interpretatbility of DNN modules. PLL regulates that each training instance is assigned a set of candidate labels that include the true label, and we were curious about ***how CAM changes in this case***. Thus, we ***for the first time*** implemented the pilot experiments in ***Figure 1*** and ***Appendix B.7***. According to these experiments, we observed two important features during the training phase, namely power and dynamic. As shown in ***Figure 1***, the power attribute illustrates that CAM could recognize more true labels than the normal outputs from the candidate sets during the very beginning of the training phase, motivating us to consider that the classifier may learn more accurate true labels if it is forced to approximate the label distribution recognized by CAM, which can discriminate more accurate labels than the basic outputs. The dynamic attribute shows that such superiority of CAM is synchronously changed with the capacity of the classifier. The accuracy inflection points between the classifier and CAM during the training phase appear at the similar location, which reveals that if the classifier learns the label distribution by CAM, then CAM may synchronously update itself as the classifier becomes stronger, so the classifier shall continue to learn the label distribution generated from the better CAM. In other words, such guidance of CAM is similar to EM Algorithm. The power attribute may enable the classifier to improve the performance by learning the true labels recognized from CAM, which is able to find more accurate labels than the classifier itself. The dynamic attribute may ensure its convergence since the CAM could be self-improved during the training process.
> > > > > >
> > > > > > To overcome the limited application of CAM, which could only be applied in specific network architecture and image datasets, we ***for the first time*** proposed the CAV to represent such knowledge in a general way. Specifically, CAV is not confined to the network and datasets, which could be ***insightful and meaningful*** to address PLL or even more WSL problems. Thus, we would like to ask R3 to rethink the ***importance and potential application*** of CAV instead of being tangled with the "minor modification of CAM". The pilot experiments in ***Figure 1 and Appendix B.7*** also show that our CAV possesses the same attributes. Although these empirical results could not serve as strict mathematical support, we have done quantitative and qualitative experiments to validate the effectiveness and superiority of our CAVL. The implementation of CAV and CAVL is ***extremely understandable and intuitive***, and their success also validates the rightness of our pilot experiments, so we contend that it is exciting and insightful to find this simple but novel approach to address this WSL problem. The CAVL is an inspiring exploration to solve WSL problem by reasonably utilizing the internal knowledge obtained from the network itself, which may reveal that the network might learn more than we thought. Thus, we would like to sincerely ask ***R3*** to re-evaluate our work, which is ***not supposed to be rejected (rating 3)*** simply due to the lacking of strict theoretical support. In other words, we strongly hope that ***R3*** could
> > > > > > understand and attach more attention to our contribution including but not limited to ***the initiative and interesting findings of the CAM, the intuitive implementation and potential values of our CAV and CAVL***.

---

### Official Review · Reviewer_Dgv3 · 2021-10-29

**Correctness:** 4
**Technical Novelty And Significance:** 3
**Empirical Novelty And Significance:** 3
**Recommendation:** 8
**Confidence:** 4

**Main Review:**

Pros:

(1) Interesting empirical observation. CAM is well known as a simple technique for discriminating the learning patterns of classes. This paper shows that it can be also used for selecting the true label from candidate labels by calculating the number of positive elements in the CAM of candidate labels.

(2) Novel methodological contribution. This paper proposes Class Activation Value (CAV, which is versatile in models and inputs) and uses CAV to select the true label from candidate labels. In this way, a novel partial label learning method is obtained. This method is novel while keeping its simplicity.

(3) Experiments are extensive and validate the effectiveness of the proposed method.

Cons:

(1) I understand that CAM can be used for selecting the true label from candidate labels by calculating the number of positive elements in the CAM of candidate labels. However, I feel that it took me a long time to understand this operation. There is only one sentence about that. So the authors are encouraged to give more descriptions on how we use CAM to select the true label. This will make it easier for readers to understand.

Questions:

(1) I feel that this paper actually focuses on the pseudo labeling task in a weakly supervised learning problem. So I would like to how can the proposed method be used in another weakly supervised learning problem like semi-supervised learning. Can the authors provide some discussions on that?

(2) The authors mentioned the average-based strategy and the identification-based strategy for partial-label learning. I think the proposed method in this paper is an identification-based strategy, and the authors are encouraged to clearly state that in this paper.



**Summary Of The Paper:**

This paper studies an interesting weakly supervised learning problem called partial-label learning, which solves the multi-class classification problem, where each training instance is assigned a set of candidate labels that include a set of candidate labels that include the true label. This paper empirically shows that the Class Activation Map (CAM), a simple technique for discriminating the learning patterns of each class in images, is better at making accurate predictions than the model itself on selecting the true label from candidate labels. Based on this motivation, this paper proposes a versatile version of CAM called Class Activation Value (CAV) and proposes a novel method to selects the true label by the maximum CAV for partial-label learning. Experiments validate the effectiveness of the proposed method.

**Summary Of The Review:**

This paper investigates partial-label learning. This paper makes two key contributions including an interesting empirical observation and a novel methodological contribution. To the best of my knowledge, the empirical observation is disclosed by this paper for the first time and the proposed method is simple yet effective. Comprehensive experimental results support what the authors claimed in the paper.

---

> ### Author Response · Authors · 2021-11-14
> **Response to Reviewer Dgv3 (R2)**
>
> Thanks for your positive feedback and constructive suggestions. We are encouraged that you contend that our findings and method are interesting and novel. Below are the responses to your comments/questions.
>
> ***
> **Q1: The process of selecting true labels from CAM.**
>
> **A1**: Thanks for your sincere suggestion! From Eq.(2), it is obviously CAM of input $x$ related to the $j$-th class, namely $m^j$, is actually a matrix. To obtain elements that are useful for representing the $j$-th class in $m^j$, we treat all the elements with positive values as the foreground seeds and ignore those elements with negative values [8-9]. In this way, we calculate the number of the positive elements in $m^j$ to represent the tendency of CAM to the $j$-th class. In fact, $x$ could have $k$ CAMs since there are $k$ classes in total. Note that the ground truth is always in the candidate set, thus we could simply select the label whose CAM has the maximum number of foreground seeds from the candidate set as the true label. We have added this discussion in ***Section 2.2*** for better understanding to the readers.
> ***
>
> **Q2: Application in other weakly supervised learning problems.**
>
> **A2**: Thank you for coming up with this insightful comment! The success of our CAVL validates the possibility of solving PLL problem by ***exploring the internal knowledge*** made from the classifier itself, which could be a new perspective for researchers to solve weakly supervised learning (WSL) problems since we may construct ***useful and reliable labels without massive human annotations***. For semi-supervised learning, we have a small number of labeled data and a large number of unlabeled data. Exploring the internal knowledge may help the classifier deal with those unlabeled datasets. Specifically, it is possible to take advantage of CAM and the normal outputs from the well-trained classifier, which is trained from those samples with complete labels, to guide the classifier to learn from useful information from those unlabeled datasets. For noisy-label learning, it is also practical to explore how CAM related to the annotated label is affected during the training phase, which may help find or design new internal knowledge for the classifier to approach the ground truth. On the other hand, we could find different forms of internal knowledge for addressing various WSL problems. As mentioned in this paper, the ***network might learn more than we thought***. WSL aims to liberate humans from exhaustive annotation work, and we believe that ***exploring the internal knowledge of the model to identify the true labels*** may be an efficient and promising method to solve other WSL problems. Therefore, we will continue to investigate it in the future.
>
> ***
> **Q3: Missing the claim of the identification-based strategy for CAVL.**
>
> **A3**: We sincerely thank the reviewer for raising this concern and we have added it in ***Section 5***.

---

> > ### Comment · Reviewer_Dgv3 · 2021-11-23
> > **My questions have been addressed.**
> >
> > Thanks for the authors' reply. My questions have been well addressed, thus I would like to keep my original score and vote for acceptance for this paper.

---

> > > ### Author Response · Authors · 2021-11-23
> > > **Response to Reviewer Dgv3 (R2)**
> > >
> > > Thanks sincerely for your positive evaluation of our work, and we are very grateful that you kept your original score and voted for acceptance, which is a great encouragement to us.

---

### Official Review · Reviewer_1wbR · 2021-11-03

**Correctness:** 3
**Technical Novelty And Significance:** 2
**Empirical Novelty And Significance:** 3
**Recommendation:** 6
**Confidence:** 5

**Main Review:**

1. It is innovative to use CAM to select true labels from candidates. The paper provides a new perspective to transfer PLL problems to supervised learning, which is worth deeper research in future work.
2. The proposed CAVL method is effective and efficient.
3. The experiments are sufficient, including various types of data and backbone. The ablation study on partial label generation methods strongly supports the intuition that CAM does not rely on any assumption on the collected data as previous work does.


weaknesses：
1. The main contribution is to utilize CAM to solve PLL problems and make some modifications on CAM to adapt to more data types and backbone. The original contributions do not seem to be enough and the novelty is a little weak.
2. The explanation of the power and dynamic attributes is not clear. Referring to section 2.2, how does these two attributes show that CAM can act as the guidance for PLL problems.
3. In the CAVL method, referring to section 3.2, the potential true label is selected based on CAV after training the model for only one epoch instead of several epochs. How do you decide when to select the potential true label?  What if the wrong labels were chosen, I did not see experiments or explanations in this case.
4. The paper needs to also provide the results and notice that realistic dataset instead of using generated visual partial labels only.
e.g. the datasets and references should be considered in the related work survey or experiments:
Partial Label Learning via Label Enhancement AAAI 19
General Partial Label Learning via Dual Bipartite Graph Autoencoder AAAI 20
GM-PLL: Graph Matching based Partial Label Learning TKDE 21

questions:
The CAV is similar to adding attention values, what are the most important novelty and differences of the method with other closely related work.


**Summary Of The Paper:**

This paper intends to exploit the learned representation of the model to tackle partial-label learning tasks. The paper begins by a PILOT experiment to show that the class activation map is better at selecting the true label from the candidate set than the model output itself. To overcome the limitation that CAM cannot be applied to linear model and non-image data, they propose class activation value (CAV) to replace CAM and show similar properties.

**Summary Of The Review:**

The paper proposes a novel CAV-based method to solve PLL problems and shows a promising performance than several state-of-the-art methods. Based on CAV, they propose a CAV learning method to select a potential true label for training and thus transfer PLL to supervised learning. Extensive and solid experiments prove the effectiveness of the proposed method. The paper provides a new way to tackle the PLL problem. The  main concern is the CAV is similar to the attention mechanism, and some of paragraphs are still need to explain.

---

> ### Author Response · Authors · 2021-11-14
> **Response to Reviewer 1wbR (R1) (PART 1/2)**
>
> Thanks for your valuable comments, and we are glad that you find it is novel that the CAM could be a guidance for purifying the true labels from the candidate sets. Below are the responses to your comments/questions.
>
> ****
> Q1. The main contribution and the novelty.
>
> A1. Thanks for raising this concern! We would like to emphasize that our major contribution actually
> is the ***finding of the positive effect of CAM on addressing PLL***, which is also acknowledged by you.
> To the best of our knowledge, this paper for the first time finds that CAM could be used to solve the PLL
> problem. Beyond the simple attention operation, CAM is essentially the exploration of the learning
> representation for a CNN-based classifier. This simple tool represents the knowledge owned by the network.
> The pilot experiments in ***Figure 1*** show that CAM is able to recognize more ground-true labels
> than the basic model outputs at the first beginning of the training epoch,
> which manifests that the CNN module may work potentially better than we thought.
> Inspired by such results, we are motivated to explore new internal knowledge in simple or shallow modules
> such as Linear and MLP where CAM is confined to.
>
> Our second contribution is the proposed CAV and CAVL method.
> Here we respectfully ***disagree that CAV is similar to normal attention modules***. We argue that
> the aim of normal attention such as SENET [1] and CBAM [2] is to ***augment the information in DNN***,
> which is beneficial to the network during the training phase. For instance, Woo *et al*. [2] proposed
> the channel attention and spatial attention to augment the context information of the feature maps.
> However, CAM and CAV represent the ***internal knowledge of the network***, and we functionalize them to
> infer the true labels from the candidate sets without human annotation.
> Additionally, it is surprisingly found that ***CAV is similar to the gradient of the Softmax function
> with respect to the input***. Thus we believe it is meaningful for the propose of CAV due to such coincidence.
> Thanks to the proposed CAV, we successfully turn PLL into a simple supervised learning problem,
> and the effectiveness of the CAVL also shows the success of exploring internal knowledge for PLL.
> The conclusion could be enlightening to weakly supervised learning (WSL),
> where many researchers have been working hard to model the loss function by imposing different assumptions
> on the data distribution. ***By exploring the internal knowledge of the model to construct reliable labels***,
> it is possible to turn many WSL problems such as semi-supervised learning and noisy-label learning
> into a simple supervised learning problem ***without much limitations on models and data distribution***.
> We will continue to investigate it in future work.
> ****
> Q2. Explanation of dynamic and power.
>
> A2. The dynamic and power attributes are two observed features for CAM during the pilot experiments.
> As shown in ***Figure 1***, the power attribute illustrates that CAM could recognize more true labels than the
> normal outputs from the candidate sets during the very beginning of the training phase,
> motivating us to consider that the classifier may learn more accurate true labels
> if it is forced to approximate the label distribution recognized by CAM, which can discriminate
> more accurate labels than the basic outputs. The dynamic attribute shows that such superiority of CAM is
> synchronously changed with the capacity of the classifier. The accuracy inflection points between the
> classifier and CAM during the training phase appear at similar location, which reveals that if the
> classifier learns the label distribution by CAM, then CAM may be synchronously self-improved
> as the classifier becomes stronger, so the classifier shall continue to learn
> the label distribution generated from the better CAM. In other words, such guidance of CAM
> is similar to EM Algorithm. The power attribute may enable the classifier to
> improve the performance by learning the true labels recognized from CAM,
> which is able to find more accurate labels than the classifier itself.
> The dynamic attribute may ensure its convergence since the CAM could be self-improved
> during the training process. Due to the limited application of CAM, we designed the CAV as its substitution.
> The experimental results in ***Section 4*** also demonstrated the rightness
> and effectiveness of our motivation. Thanks to the reviewer for this constructive suggestion and
> we have added a more detailed description of these two attributes in ***Section 2.2***.

---

> > ### Author Response · Authors · 2021-11-14
> > **Response to Reviewer 1wbR (R1) (PART 2/2)**
> >
> > **Q3: Influence of epoch settings and false positive labels on CAVL.**
> >
> > **A3**: Thanks for your constructive comments! We added the relevant experiments in ***Appendix B.3***,
> > the results show that our CAVL becomes less effective with the increase of the starting epoch
> > for selecting the maximum CAV as the true label. The classifier is negatively affected by the
> > false positive labels with long time training by IM, which simply regards all the labels
> > from the candidate sets as the true label with using cross entropy loss. This phenomenon
> > is reasonable because the DNN learn patterns first, which suggests that deep networks can
> > gradually memorize the data, moving from regular data to irregular data such as outliers
> > and mislabeled data [3]. Therefore, the larger starting epoch means the less training time for our CAVL,
> > resulting in more difficulty for CAVL to address this memorization problem during the rest of the training phase.
> >
> > We ***agree*** with you that CAV possibly selects the false positive labels for the classifier,
> > while we should note that ***all existing IBS-based approaches suffer from this common problem***
> > since there is no way to ideally identify the true label for every instance in PLL. Wen *et al*.
> > [4] has stated in Related Work that “... it is worth highlighting that these algorithms have
> > shown their weaknesses when facing the false positive labels...”. Yao *et al*. [5] has stated
> > in Related Work that “... One potential shortcoming of identification-based methods is that
> > the identified label in the current iteration may turn out to be false positive...”.
> > According to the pilot experiments in ***Figure 2*** and ***Appendix B.7***,
> > our CAV is obviously able to select more true labels than the classifier itself.
> > Furthermore, the experimental results in ***Section 4*** also sufficiently validates the effectiveness
> > and superiority of our CAVL, which is also mentioned in your comment about the strength to CAVL.
> > Therefore, we believe that our CAVL could well relieve the problems caused by the false positive labels.
> >
> > On the other hand, we would like to emphasis that ***ABS also could not possibly alleviate the negative
> > effects of the false positive labels*** since all labels are considered without differentiation
> > during the training phase. Feng *et al*. [6] stated in Section 2 that “...the obvious drawback is
> > that ground-truth label may be overwhelmed by other candidate (false positive) labels...”. Yan *et al*. [7] stated in Related Work that “...The simple average-based strategy however
> > fails to explore the difference among the candidate labels...”. Therefore, we contend that
> > negative influence from the false positive labels is a ***common challenge in PLL***.
> > ****
> > **Q4: Missing references and results.**
> >
> > **A4**: Thanks for your sincere suggestion and we added those references in ***Section 5***.
> > We would like to remind the reviewer that we have reported the experimental results on real-world
> > datasets in ***Table 3***. In ***Section 4.1.2***,
> > we also stated that we used the given candidate labels of the real-world datasets
> > ***without labels generated by FPS or USS*** (see the line “...Note that these real-world datasets have their candidate labels...”).

---

> ### Comment · Reviewer_1wbR · 2021-11-20
> **Further Discussion**
>
> Thanks for the responses from the authors. The responses solved all of my questions or concerns in Q2 and Q4. There are remaining issues about the responses for Q1 or Q3.
>
> About Q1:
> From experiment results, the usage of the CAM (previously proposed class activation map) increases the performances of the PPL models. But my question cares about the novel ideas of the original contributions of the methods or technologies. Author responses claim the CAM "represents the knowledge owned by the network" and "recognize more ground-true labels than the basic model outputs at the first beginning of the training epoch, which manifests that the CNN module may work potentially better than we thought". These descriptions are still vague and unclear. What types of knowledge (only visual attentions or others, What if the task is not visual recognition tasks)? How the CAM represents knowledge in your method, is that novel? The author responses about CAV claim the novelty of the CAM and CAV are "represent the internal knowledge of the network, and we functionalize them to infer the true labels from the candidate sets without human annotation", "CAV is similar to the gradient of the Softmax function with respect to the input". From this description,  it seems still vague and the claimed novelty is still the usage of CAM or CAV instead of the method.
>
> About Q3:
> Author responses agree that the CAM or CAV would "possibly select the false positive labels for the classifier" and all similar methods "suffer from" the false positive issue. I suggest adding these points in the conclusions so that the effective ranges of the CAM or CAV could be clarified.
> The responses also mentioned, "CAV is obviously able to select more true labels than the classifier itself". So what types of errors that the CAV would alleviate for PLL? If the paper could add the discussions of the results, it would be more objective.

---

> > ### Author Response · Authors · 2021-11-21
> > **Response to Reviewer 1wbR (R1) (PART 1/2)**
> >
> > **Q1**. More explanation about our method.
> >
> > **A1**. Thanks for raising these concerns. Here we firstly provide a detailed explanation of CAM. For a well-trained CNN module for addressing a classification problem, the Class Activation Map (CAM) is simply a weighted linear sum of the presence of the visual patterns at different spatial locations for the input images. CAM aims to dig out ***what regions that the network focuses on the input images related to true labels***. It is known that a CNN-based module is a "black box" and the meaning behind CAM is the exploration of the working mechanism inside such technology. Based on CAM, it is convenient to visualize the learning patterns of the CNN-based modules. We added several samples of CAM generated from [8] in ***Appendix C***. Here we take one example of the Teapot from ***Figure 4*** for illustration, the CAM of the image related to the teapot shows the most discriminative regions in the head part, which manifests that the classifier classifies the image to the teapot since it finds and concentrates on the head part of the teapot. Thanks to CAM, we could conveniently and effectively investigate the intrinsic principles of CNN-based modules by simply utilizing the components from the network itself. This is also the reason why we name it "internal knowledge". Obviously, we here would like to emphasize again that ***CAM is generally confined to CNN-based modules with image inputs.***
> >
> > PLL regulates that each training instance is assigned a set of candidate labels that include the true label, and we were curious about ***how CAM changes in this case***. Thus, we ***for the first time*** implemented the pilot experiments in ***Figure 1*** and ***Appendix B.7***, and we found that CAM could address PLL due to the observed attributes, namely power and dynamic. Observing these attributes in CAM, we were reasonably and intuitively eager to solve PLL by using CAM as the guidance. Here we would like to emphasize that this is ***the first time*** that it is possible to solve one of the WSL problems by simply using internal knowledge. Unfortunately, CAM is merely proposed on the CNN-based backbones to address image inputs, limiting its application to a large extent. This is also the reason and motivation that why we try to seek a substitution for CAM to improve its generalization. Thus, we proposed the CAV, and the pilot experiments had validated that our CAV showed the same attributes as the CAM. Inspired by CAM, CAV essentially represents the focus of the network to the true labels in a more general way. Intuitively, we proposed the CAVL to address the PLL problem by simply treating the CAV as the guidance. The implementation of CAV and CAVL is ***extremely understandable and intuitive***, and their success also validates the rightness of our pilot experiments, so we contend that it is exciting and insightful to find this simple but novel approach to address this WSL problem. Surprisingly, we also find that the implementation of CAV happens to be the gradient of the Softmax function with respect to the input, which is an interesting finding that may inspire us and others to find the intrinsic principles in the future. We believe that the above findings and the success of our CAVL could reveal that ***the network might learn more than we thought***, which is significant to solve more WSL problems by finding reliable labels from the network itself. In all, we would like to ask ***R1*** to attach more attention to our contribution including but not limited to ***the initiative and interesting findings of the CAM, the intuitive implementation and potential values of our CAV and CAVL***.

---

> > ### Author Response · Authors · 2021-11-21
> > **Response to Reviewer 1wbR (R1) (PART 2/2)**
> >
> > **Q2**. Discussion on the effective ranges of our methods and types of errors.
> >
> > **A2**. Thanks for your comments. Here we would like to emphasize that we have stated that our CAVL could potentially select false positive labels in ***Section 5*** following the suggestion by ***R2***.
> >
> > For the discussion of types of errors in CAVL, we would like to respectfully remind you that there are ***no clear definitions for the types of errors in PLL*** and there are ***no existing studies in PLL [4-7,14-17]*** that include such a somewhat strange analysis. Different from CAM, our CAV could address various forms of input datasets by using either deep or shallow neural networks as the backbones. The pilot experimental results in ***Figure 1*** and ***Appendix B.7*** also validated that the findings of CAM or CAV do not rely on any assumption on the collected data, and CAV or CAM could select more true labels (less false positive labels) than the normal outputs, which also received your appreciation. The influence of the epoch settings on CAVL indeed could be universal, but we could not figure out ***what common errors could be defined in PLL*** with different assumptions and input forms on the collected data, and we contend that this is also the reason why none of existing work in PLL provided this discussion, which seems to be trivial and meaningless.

---

> > ### Comment · Reviewer_1wbR · 2021-11-22
> > **Comment Summary**
> >
> > Overall Summary: Thanks for the detailed explanations of the author's responses.  Considering the response efforts, I raised the review score. But the paper and responses did make several important confusions or mistakes that need to be clarified or corrected (please refer to the detailed comments below). So the recommendation is "marginally over the acceptance threshold". The paper should correct the usage of the terminology "internal knowledge" and provide necessary discussions about the Q2.
> >
> > About the response to Q1:
> > The response to Q1 confused the concepts of "visual representation" and "knowledge of NN". In fact, CAM is the visual representation, which is calculated by a weighted sum of the spatial feature maps of the last convolutional layer. CAM is a form of intermediate outputs instead of knowledge of NNs. The concepts should be used with caution instead of naming them without considering their original usage no matter the paper is accepted or not.
> >
> > About the response to Q3:
> > The "false-positive errors" are very common issues in machine learning and semi-supervised learning, especially in PLL problems. Since the true labels may be overwhelmed by the false positive labels in PLL problem settings. Please refer to recent papers [1-3], this problem is being widely discussed.
> > So the opinions of the last responses to Q3 seem to be totally wrong: "no clear definitions for the types of errors in PLL" and "there are no existing studies in PLL".
> >
> > [1] Progressive Identification of True Labels for Partial-Label Learning, PMLR 20
> > [2] Partial Label Learning via Feature-Aware Disambiguation, IJCAI 18
> > [3] Solving the Partial Label Learning Problem: An Instance-based Approach, IJCAI 15

---

> > > ### Author Response · Authors · 2021-11-23
> > > **Response to Reviewer 1wbR (R1)**
> > >
> > > Thanks sincerely for your great efforts in our work and we are very grateful that you raised the score. Below are our responses to your new comments.
> > > ***
> > > **Q1**. Claims about CAM.
> > >
> > > **A1**. Thank you for this constructive suggestion and we respectfully agree with your comment. Therefore, we revised our claim of CAM as the ***"internal representation", which essentially is a form of intermediate outputs.*** We have made relevant modifications related to "internal knowledge" in our latest manuscript.
> > > ***
> > > **Q2**. Discussion on the false positive labels.
> > >
> > > **A2**. Thanks for your comments. We would like to ***express our confusion*** about your previous comments that "what types of errors" should be considered. As you provided three papers [1-3], ***we would like to humbly ask R1 to clearly point out ”what types of errors” are defined in these three papers, and we will definitely provide any experiments or analyses if there are some clear statements about what types of analyses and experiments should be supplemented.***
> > >
> > > To the best of our knowledge, the three papers have indeed commonly conducted ablation studies related to the false positive labels by manually controlling the number of false positive labels in the candidate set. Specifically, these three papers all utilized a flipping probability $q$, which defines the probability that an irrelevant label enters the candidate label set (becomes a false positive label). Specifically, the number of false positive labels is fixed in [2-3] (the maximum number is 3). Following similar settings, we also provided relevant experimental results in ***Table 2***. Note that we directly set $q$ to all irrelevant labels (please refer to ***Appendix B.4***), which would be more difficult for the models to recognize the true labels. The experimental results in ***Table 2*** illustrated that our model became more vulnerable with the increase of potential false positive labels, achieving the lowest performance when $q=0.7$. Besides, we conducted additional experiments to show how our CAVL performed when facing the false positive labels generated by USS in ***Table 1***. Therefore, we believe that we have investigated the effects of false positive labels on our model through different values of $q$, and ***we firmly contend that the above experiments in [1-3] do not include the discussion about "types of errors/false labels" since they did not clearly define or discuss such type***.

---

> > > > ### Comment · Reviewer_1wbR · 2021-11-29
> > > > **Comments about Q3**
> > > >
> > > > There might be a misunderstanding, all key points of the Q3 are about the "false positive" issue instead of "error type definition" in PLL. And the potential issue has widely been discussed in the PLL domain (e.g. in the three mentioned references).
> > > >
> > > > Previous authors' responses tried to explain that the proposed method "possibly select the false positive labels for the classifier" and was also be "suffer from" the false positive issue, but the result analysis in another response also claims that "CAV or CAM could select more true labels (less false positive labels) than the normal outputs". Although these two claims exist partial differences, the previous author's responses basically provided the answers to the question. So all my questions are answered, thank you.

---

> > > > > ### Author Response · Authors · 2021-11-30
> > > > > **Response to Reviewer 1wbR (R1)**
> > > > >
> > > > > Thanks for your reply and we are glad that all your questions are answered to better understand our contribution according to our rebuttal. Please let us know if you have any other concerns and we will definitely provide detailed explanations.

---

### Author Response · Authors · 2021-11-14
**Summary of our Rebuttal**

***(Reviewer 1wbR --- ***R1***, Reviewer Dgv3 --- ***R2***, Reviewer sUUe --- ***R3***)***.

We sincerely appreciate all reviewers for their great efforts in providing many treasurable and constructive comments.
We are encouraged that they consider our CAVL is novel (***R2***), effective and efficient (***R1***). Besides, We are glad that all reviewers agree that it is a strength to use CAM to solve PLL. We promise that our source code will be released once our paper is accepted. Below are some key changes to our manuscript:

1.	We have provided the discussion on ***the epoch setting*** in our CAVL in ***Appendix B.3*** (suggested by ***R1***).
2.	We have added some ***relevant references***, and pointed out that ***our method belongs
to IBS*** in ***Section 5*** (suggested by ***R1*** and ***R2***).
3.  We have added more detailed discussions in ***Section 2.2*** on ***how to select true labels using CAM
in Figure 1***, and ***how to solve PLL using two observed attributes of CAM*** (suggested by ***R1*** and ***R2***).
4.	We have provided more ***pilot experiments on real-world datasets*** in ***Appendix B.7***,
and the relevant experimental results of ***PRODEN*** (suggested by ***R3***).
5. We have added a detailed illustration for CAM in ***Appendix C***.
6. We have modified some claims about CAM and renamed it as ***"internal representation"***. (suggested by ***R1***).

In addition, we find that there are ***three main concerns*** raised by all reviewers:

1) The first concern is the ***novelty of the CAV*** and its ***potential application in future work***. The corresponding responses can be found in A1 to ***R1*** and A2 to ***R2***.
2) The second concern is the ***influence of positive labels***, and the corresponding responses can be found in A3 to ***R1*** and A2 to ***R3***.
3) The third concern is the ***experimental settings***, and the corresponding responses can be found in A1 and A5 to ***R3***.

Finally, we list the references below that are mentioned in our responses:

[1] Squeeze-and-Excitation Networks, TPAMI, 2020.
[2] CBAM: Convolutional Block Attention Module, ECCV, 2018.
[3] A closer look at memorization in deep networks, ICML, 2017.
[4] Leveraged Weighted Loss for Partial Label Learning, ICML, 2021.
[5] Deep Discriminative CNN with Temporal Ensembling for Ambiguously-Labeled Image Classification, AAAI, 2020.
[6] Leveraging Latent Label Distributions for Partial Label Learning, IJCAI, 2018.
[7] Partial Label Learning with Batch Label Correction, AAAI, 2020.
[8] Learning Deep Features for Discriminative Localization, CVPR, 2016.
[9] Grad-CAM: Visual Explanations from Deep Networks via Gradient-based Localization, ICCV, 2017.
[10] Progressive Identification of True Labels for Partial-Label Learning, ICML, 2020.
[11] Efficientnet: Rethinking model scaling for convolutional neural networks, ICML, 2019.
[12] Interpretable explanations of black boxes by meaningful perturbation, ICCV, 2017.
[13] Self-supervised equivariant attention mechanism for weakly supervised semantic segmentation, CVPR, 2020.
[14] Network Cooperation with Progressive Disambiguation for Partial Label Learning, ECML, 2020.
[15] Provably Consistent Partial-Label Learning, NIPS, 2020.
[16] Partial Label Learning with Unlabeled Data, IJCAI, 2019.
[17] Adaptive Graph Guided Disambiguation for Partial Label Learning, TPAMI, 2021.

---

### Decision · Program_Chairs · 2022-01-20

**Decision:**

Accept (Poster)

**Comment:**

This paper considers the so-called partial-label learning problem and proposes a class activation map that is better at making accurate predictions than the model itself on selecting the true label from candidate labels. The authors investigate the approach in experimental results on four benchmark image datasets.

The reviewers appreciated the simplicity of the approach and its effectiveness in practice. The reviewers raised questions how to apply the approach to another weakly supervised learning problem such as semi-supervised learning and whether the approach is an identification-based strategy. The reviewers also raised several questions asking for more details.

The authors submitted responses to the reviewers' comments. After reading the response, updating the reviews, and discussion, the reviewers who took part in the discussion considered that their  “questions have been well addressed” and that the “authors’ responses basically provided the answers to the questions”.

The feedback provided was already fruitful and the final version should be already improved.

Accept. Poster.